

# Impacts of heterogeneous uptake of dinitrogen pentoxide and chlorine activation on ozone and reactive nitrogen partitioning: Improvement and application of WRF-Chem model in southern China

Qinyi LI[1], Li ZHANG[1], Tao WANG[1]*, Yee Jun THAM[1], Ravan AHMADOV[2,3], Likun XUE[4], Qiang ZHANG[5], and Junyu ZHENG[6]

[1] Department of Civil and Environmental Engineering, The Hong Kong Polytechnic University, Hong Kong, China,
[2] Cooperative Institute for Research in Environmental Sciences, University of Colorado at Boulder, Boulder, CO, USA,
[3] Earth System Research Laboratory, National Oceanic and Atmospheric Administration, Boulder, CO, USA,
[4] Environment Research Institute, Shandong University, Ji'nan, China,
[5] Center for Earth System Science, Tsinghua University, Beijing, China,
[6] School of Environmental Science and Engineering, South China University of Technology, Guangzhou, China.

* *Correspondence to:* T. Wang (cetwang@polyu.edu.hk)

**Abstract**: The uptake of dinitrogen pentoxide ($N_2O_5$) on aerosol surfaces and the subsequent production of nitryl chloride ($ClNO_2$) can have significant impact on the oxidising capability and thus on secondary pollutants such as ozone. The range of such impact, however, has not well been quantified in different geographical regions. In this study, we applied Weather Research and Forecasting coupled with Chemistry (WRF-Chem) model to investigate the impact of the $N_2O_5$ uptake processes in the Hong Kong-Pearl River Delta (HK-PRD) region, where the highest ever-reported $N_2O_5$ and $ClNO_2$ concentrations were observed in our recent field study. We first incorporated into the WRF-Chem an aerosol thermodynamics model (ISORROPIA II), recent parameterisations for $N_2O_5$ heterogeneous uptake and $ClNO_2$ production and gas-phase chlorine chemistry. The revised model was then used to simulate the spatiotemporal distribution of $N_2O_5$ and $ClNO_2$ over the HK-PRD region and the impact of $N_2O_5$ uptake and Cl activation on ozone and reactive nitrogen in the planetary boundary layer (PBL). The updated model is capable of reproducing the temporal patterns of $N_2O_5$ and $ClNO_2$ observed at a mountain-top site in Hong Kong, but overestimates $N_2O_5$ uptake and $ClNO_2$ production. The model results suggest that under average meteorological conditions, elevated levels of $ClNO_2$ (>0.25 ppb within the PBL) are present in the south-western PRD, with the highest values (>1.00 ppb) predicted near the ground surface (0-200 m above ground level (a.g.l.)). In contrast, during the night when very high levels of $ClNO_2$ and $N_2O_5$ were measured in well-processed plumes from the PRD, $ClNO_2$ is mostly concentrated within the residual layer (~300 m a.g.l.). The addition of $N_2O_5$ heterogeneous uptake and Cl activation reduces the NO and $NO_2$ levels by as much as 1.93 ppb (~7.4%) and 4.73 ppb (~16.2%), respectively, increases the total nitrate and ozone concentrations by up to 13.45 µg m$^{-3}$ (~ 57.4%) and 7.23 ppb (~16.3%), respectively, in the PBL. Sensitivity tests show that the simulated chloride and $ClNO_2$ concentrations are highly sensitive to chlorine emission. Our study suggests the need to measure the vertical profiles of $N_2O_5/ClNO_2$ under various meteorological conditions, to consider the chemistry of $N_2O_5/ClNO_2$ in the chemical transport model, and to develop an updated chlorine emission inventory over China.





## 1 Introduction

Dinitrogen pentoxide ($N_2O_5$) is mostly produced by chemical reactions involving ozone ($O_3$) and nitrogen dioxide ($NO_2$).

$$O_3 + NO_2 \longrightarrow NO_3 \qquad (1)$$

$$NO_3 + NO_2 \longrightarrow N_2O_5 \qquad (2)$$

The subsequent heterogeneous uptake of $N_2O_5$ produces nitrate on water-containing aerosol surfaces via reaction 3 (hydrolysis) and produces both nitrate and gaseous nitryl chloride ($ClNO_2$) on chloride-containing aerosol surfaces via reaction 4 (Finlayson-Pitts et al., 1989; Osthoff et al., 2008). The net reaction of reactions 3 and 4 could be treated as reaction 5, in which the $ClNO_2$ yield, i.e. parameter $\phi$, represents the fraction of $N_2O_5$ that reacts via reaction 4. The produced $ClNO_2$ can be further photolysed into Cl radical and $NO_2$ (via reaction 6).

$$N_2O_5(g) + H_2O(aq) \longrightarrow 2\ HNO_3(aq) \qquad (3)$$

$$N_2O_5(g) + HCl(aq) \longrightarrow HNO_3(aq) + ClNO_2(g) \qquad (4)$$

$$N_2O_5(g) + (1-\phi)H_2O(aq) + \phi\ HCl(aq) \longrightarrow (1-\phi)\times 2\ HNO_3(aq) + \phi\times(HNO_3(aq) + ClNO_2(g)) \qquad (5)$$

$$ClNO_2(g) + h\nu \longrightarrow Cl\,(g) + NO_2(g) \qquad (6)$$

The above processes affect the fate and composition of the total reactive nitrogen ($NO_y$), which is the sum of NO, $NO_2$,
$HNO_3$ (g), $2*N_2O_5$, $NO_3$, $ClNO_2$, PAN, HONO, $HNO_4$, aerosol nitrate, and various organic nitrates. The hydrolysis of $N_2O_5$ is the major loss pathway for $NO_x$ (=NO+$NO_2$) at night, reducing the amount of $NO_x$ for daytime photochemistry in the following day, while producing nitrate aerosol contributing to secondary aerosol (Brown and Stutz, 2012). When $ClNO_2$ is produced, it serves as a reservoir for reactive nitrogen at night and is photolysed to recycle $NO_2$ and release highly reactive chlorine radicals (Cl activation), both of which can significantly affect the daytime photochemistry, such as $O_3$ formation via reactions with
volatile organic compounds (VOCs) (Atkinson, 2000; Thornton et al., 2010; Riedel et al., 2014).

The critical parameters required to determine the impacts of the $N_2O_5$ uptake processes are the rate constant of reaction 5, $k_5$, and the yield of $ClNO_2$, $\phi$. $k_5$ can be calculated from Eq. (1) by treating the $N_2O_5$ heterogeneous uptake reaction as a first order reaction (Chang et al., 2011),

$$k_5 = \frac{V_{N_2O_5} \times S_{aer} \times \gamma}{4}, \qquad \text{(Equation 1)}$$

where $V_{N_2O_5}$ denotes the mean molecular velocity of $N_2O_5$, $S_{aer}$ is the aerosol surface to volume ratio, and $\gamma$ represents the heterogeneous uptake coefficient of $N_2O_5$. $V_{N_2O_5}$ and $S_{aer}$ are relatively well determined; therefore, the treatments of $\gamma$ and $\phi$ are crucial for the prediction of the impacts of $N_2O_5$ uptake and Cl activation. In terms of $\gamma$, a fixed value of 0.1 was first proposed (Dentener and Crutzen, 1993). Later studies considered dependence of $\gamma$ on the aerosol species/compositions (sea



salt, black carbon, sulfate, nitrate, chloride, organic matter and water), temperature and/or relative humidity (Evans and Jacob, 2005 and the reference therein; Davis et al., 2008; Anttila et al., 2006; Riemer et al., 2009; Bertram and Thornton, 2009). Several parameterisations have been proposed for the yield of $ClNO_2$. Simon et al. (2010) applied a constant value of 0.75 for the fraction of $N_2O_5$ involved in the production of $ClNO_2$. More detailed parameterisations of $\phi$ considering the effects of

aerosol compositions were proposed by Roberts et al. (2009) and by Bertram and Thornton (2009).

Several studies have been conducted to examine the impacts of $N_2O_5$ uptake *or* $ClNO_2$ production with the use of the chemical transport model. Lowe et al. (2015) and Archer-Nicholls et al. (2014) incorporated the heterogeneous uptake of $N_2O_5$ on particles into the MOSAIC aerosol module in WRF-Chem based on the methods suggested by Bertram and Thornton (2009), Anttila et al. (2006) and Riemer et al. (2009). Their results suggested that $N_2O_5$ uptake suppressed VOC oxidation (by OH and

$NO_3$) by a factor of 1.5 and significantly enhanced nitrate formation during nighttime (an increase from 3.5 to 4.6 $\mu g\ Kg^{-1}$) over north-western Europe. Sarwar et al. (2012) implemented the heterogeneous production of $ClNO_2$ based on the parameterisation proposed by Bertram and Thornton (2009) and additional gas-phase chlorine reactions in CMAQ and examined the impacts of Cl activation due to $ClNO_2$ production and Cl chemistry on air quality. Their results showed that $ClNO_2$ production reduced the total nitrate level (up to 0.8-2.0 $\mu g\ m^{-3}$ or 11-21%) and had modest impacts on the 8-h $O_3$

level (up to 1-2 ppb or 3-4%) in the United States. Sarwar et al. (2014) expanded the study region used in Sarwar et al. (2012) to the entire northern hemisphere and suggested that $ClNO_2$ production had remarkable impacts on the air quality in China and western Europe with enhancements of the 8-h $O_3$ level up to 7.0 ppb. Most previous studies focused on investigating the effects of $N_2O_5$ uptake or $ClNO_2$ production in North America and Europe; however, little is known about Asia. The only study that covered Asia was performed by Sarwar et al. (2014); it used a coarse model resolution (>100 Km) and considered only biomass

burning as the source of chloride over land.

Previous studies in Asia (Hong Kong) have revealed the existence and significance of $ClNO_2$ in this region (Wang et al., 2014; Tham et al., 2014; Xue et al., 2015). In a recent field study, we observed the highest ever-reported mixing ratios of $N_2O_5$ (1-minute value up to 7.7 ppbv) and $ClNO_2$ (4.7 ppbv) at a mountain top site (957m above sea level) in Hong Kong (Brown et al., 2016; Wang et al., 2016). This result indicates rapid production of $N_2O_5$ and $ClNO_2$ in the Hong Kong-Pearl River delta

(HK-PRD) region which has long suffered $O_3$ and $NO_x$ pollution (Wang et al., 2009). Meteorological analysis and chemical data revealed highly inhomogeneous dynamic and chemical processes and considerable impacts of the $ClNO_2$ chemistry on the radical budget (up to 77% increase of OH) and daytime $O_3$ production (up to 41%) in the high $ClNO_2$ air mass as it transported to downwind locations above the ocean (Wang et al., 2016). It was also suggested in that study that other locations downwind of major urban areas under prevailing conditions may experience more frequent events with high levels of

$ClNO_2/N_2O_5$ than the site in Hong Kong. It is therefore of great interest to investigate the regional distribution of $N_2O_5/ClNO_2$ and the impact of $N_2O_5$ uptake and subsequent chemistry.



This study was conducted to investigate (1) the spatial (horizontal and vertical) distribution of the $N_2O_5$ and $ClNO_2$ concentrations in the HK-PRD region and (2) the spatial extent of the impact of $N_2O_5$ uptake processes on the formation of $O_3$ and the partitioning of reactive nitrogen in this region. The latest version of a widely used aerosol thermodynamics model, ISORROPIA II (Fountoukis and Nenes, 2007), was incorporated into the MADE/VBS aerosol model. ISORROPIA II has the

ability to simulate the equilibrium between hydrogen chloride (HCl) and chloride which is critical for the simulation of $N_2O_5$ heterogeneous uptake and Cl activation. But this capacity is not considered in the current MADE/VBS model in WRF-Chem (Grell et al., 2005; Ahmadov et al., 2012). Up-to-date parameterisation for $N_2O_5$ uptake and $ClNO_2$ production and Cl radical initiated chemistry were implemented into WRF-Chem. The revised WRF-Chem was then applied to southern China to investigate the spatial distribution of $N_2O_5$ and $ClNO_2$ and the impacts of these processes on $O_3$ and $NO_y$. We start with a

description of the data used to run and validate the simulations, the amendments to the WRF-Chem model, and the model setup in Section 2. In Section 3, we show the performance of the WRF-Chem model in the simulation of several air pollutants without $N_2O_5$ uptake processes, and the simulation results of $N_2O_5$ and $ClNO_2$ with $N_2O_5$ uptake and Cl activation processes; we then evaluate the impacts of $N_2O_5$ uptake and Cl activation on $NO_x$, total nitrate, $O_3$ and $NO_y$ partitioning, and test the sensitivity of the $ClNO_2$ concentration to chlorine emissions. A summary is given in Section 4.

## 2 Methodology

### 2.1 Data

#### 2.1.1 Field measurement data

$N_2O_5$ and $ClNO_2$ concentrations were measured at Tai Mo Shan (TMS) in Hong Kong with a chemical ionization mass spectrometer between November 15 and December 5, 2013 (refer to Wang et al., 2016 for more details). The measurements

were made on a mountain top in the south-eastern PRD at an altitude of 957m. Other major air pollutants, including $PM_{2.5}$, $NO_2$, and $O_3$, were also measured at the TMS site and at 11 general (non-roadside) monitoring stations of the Hong Kong Environmental Protection Department (HKEPD; available at: http://epic.epd.gov.hk/EPICDI/air/station/). Hourly measurement data were used to validate the performance of the WRF-Chem simulations.

#### 2.1.2 Emission data

Four sets of anthropogenic emission inventories (EIs) covering different areas were adopted in this study. For mainland China, we used the Multi-resolution Emission Inventory for China (MEIC; available at http://meicmodel.org), developed by Tsinghua University for year 2010. For the PRD, the anthropogenic EI developed by the Southern China University of Technology was applied. The anthropogenic EI developed by HKEPD was used over Hong Kong. INTEX-B EI (Zhang et al., 2009) was adopted for other Asian regions. Readers are referred to Zhang et al. (2016) for the details of these anthropogenic EIs. For natural



emissions, the biogenic emission parameterisation proposed by Guenther et al. (1994), the dust emission parameterisation proposed by Shaw et al. (2008) and the sea salt emission parameterisation proposed by Gong et al. (2002) were adopted in this study. The concentrations of sodium, chloride, calcium, magnesium and potassium in dust and sea salt follow those suggested by Millero (1996) and Wedepohl (1995), as shown in Table S1.

Chlorine emissions are not included in most EIs, but they are critical for the simulation of $N_2O_5$ uptake and Cl activation. In this study, the Reactive Chlorine Emission Inventory (RCEI) (Keene et al., 1999 and references therein, available at http://eccad.sedoo.fr/eccad_extract_interface/JSF/page_login.jsf) with a resolution of $1°x1°$ was adopted to provide chlorine emissions, including emissions from biomass burning and anthropogenic activities (e.g. coal combustion). Global chlorine emissions from biomass burning and anthropogenic activities are estimated to be ~6.3 Tg Cl $yr^{-1}$ and ~6.6 Tg Cl $yr^{-1}$,
respectively. The RCEI inventory is the only available chlorine EI that currently covers China, and it is subject to some, probably large, uncertainties for representing Cl emissions in the HK-PRD region due to its low spatial resolution and the fact that it was compiled for year 1990. Coal consumption and $SO_2$ emissions from coal-fired power plants in China increased by 479% and 56% from 1990 to 2010, respectively (Liu et al., 2015); thus it is expected that chlorine emissions from coal combustion, which form a large proportion of anthropogenic chlorine (Keene et al., 1999), also increased significantly over
that period. We conducted two sensitivity simulations by adjusting the chlorine emissions to test the dependence of the chloride and $ClNO_2$ concentrations on the varied chlorine emissions; the results are shown in Section 3.5.

### 2.1.3 Meteorological data

Three-hourly meteorological measurements, including atmospheric pressure, temperature, relative humidity, wind direction and wind speed, at ~2500 surface meteorological stations and twelve-hourly data at ~250 sounding stations, were obtained
from the China Meteorological Agency (CMA) and Hong Kong Observatory and were adopted in Four-Dimensional Data Assimilation to improve the model performance of the meteorological fields using observational nudging techniques (Zhang et al., 2016). The FNL Operational Global Analysis dataset provided by the National Centers for Environmental Prediction (available at http://rda.ucar.edu/datasets/ds083.2/) was used for analysis nudging. Observational and analytical nudging techniques have been shown to improve the performance of meteorological simulation in both northern China (Zhang et al,
2015) and southern China (Zhang et al., 2016). Hourly datasets from ~500 surface meteorological observation stations obtained from the CMA were used to validate the meteorological simulations.

### 2.2 Model development

### 2.2.1 Incorporation of ISORROPIA II

ISORROPIA II, an aerosol thermodynamics model developed by Nenes et al. (1998) and Fountoukis and Nenes (2007), was
incorporated to replace the aerosol thermodynamics module in the MADE/VBS aerosol model in the original WRF-Chem v3.5





so as to extend the capacity of simulation of the equilibrium between $PM_{2.5}$ compositions and their corresponding gaseous species. The MADE/VBS model adopts volatility basis set model to simulate secondary organic aerosol (SOA) formation and provides improved simulations of SOA compared to the traditional MADE/SORGAM model (Ahmadov et al., 2012). The current MADE/VBS model only estimates the thermodynamic equilibrium between $SO_4^{2-}$, $NO_3^-$, $NH_4^+$, $H_2O$ and corresponding

gases, whereas ISORROPIA II simulates the equilibrium between $SO_4^{2-}$, $NO_3^-$, $NH_4^+$, $H_2O$, $Na^+$, $Cl^-$, $Ca^{2+}$, $Mg^{2+}$, $K^+$ and associated gases.

### 2.2.2 $N_2O_5$ heterogeneous uptake, $ClNO_2$ production and Cl gaseous reaction

We adopted the parameterisations of $N_2O_5$ heterogeneous uptake and $ClNO_2$ production suggested by Bertram and Thornton (2009). According to the parameterisations, the $N_2O_5$ heterogeneous uptake coefficient, $\gamma$, can be calculated with the following

equation:

$$\gamma = Ak\left(1 - \frac{1}{\left(\frac{0.06[H_2O(l)]}{[NO_3^-]}\right) + 1 + \left(\frac{29[Cl^-]}{[NO_3^-]}\right)}\right) \qquad \text{(Equation 2)}$$

where $A = 3.2 \times 10^{-8}$, $k = 1.15 \times 10^6 \times (1 - e^{(-0.13[H_2O(l)])})$, and $[H_2O(l)]$, $[NO_3^-]$ and $[Cl^-]$ are the molarities of liquid water, nitrate, and chloride in aerosol volume. The yield of $ClNO_2$, $\phi$, can be calculated with the following equation:

$$\phi = \left(1 + \frac{[H_2O(l)]}{483[Cl^-]}\right)^{-1} \qquad \text{(Equation 3)}.$$

The loss of $N_2O_5$ and the production of nitrate and $ClNO_2$ can be predicted with Eq. (1-3). The produced $ClNO_2$ is then photolysed, releasing a Cl atom, which further oxidises VOCs like an OH radical. The Cl-initiated gaseous chemistry used in this study was originally designed for CB05 mechanism by Sarwar et al. (2012), and was modified for RACM_ESRL mechanism (detail reactions are shown in Table S2). RACM_ESRL mechanism is the updated Regional Atmospheric Chemistry Mechanism in WRF-Chem based on the original version in Stockwell et al. (1997). The photolysis rates of $Cl_2$,

HOCl, $ClNO_2$ and formyl chloride (FMCl) were calculated with the absorption cross section and quantum yield obtained from Atkinson et al. (2007) and Atkinson et al. (2008).

We implemented the $N_2O_5$ heterogeneous uptake, the $ClNO_2$ production, and the Cl-initiated reactions into the MADE/VBS aerosol model, RACM_ESRL gas-phase mechanism and Madronich photolysis model (Madronich, 1987) in the 'RACM_SOA_VBS_KPP' chemistry option in WRF-Chem v3.5.

**2.3 Model setup**

### 2.3.1 Model configuration

The model configurations of WRF-Chem used in this study are shown in Table 1. We used the Noah model to simulate the land surface process, the YSU module to simulate the PBL processes, the Purdue Lin scheme to predict the microphysics, the



Grell 3-D ensemble module to simulate cumulus, the RRTMG model to predict shortwave and longwave radiation and the RACM_ESRL, MADE/VBS and Madronich modules to simulate gas-phase chemistry, aerosol processes and photolysis.

Model simulations were conducted in 4 domains covering East Asia, southern China, the PRD and Hong Kong, with spatial resolutions of 27, 9, 3, and 1 km, respectively (see Fig. 1a). High grid resolutions were adopted in this study to capture the extremely inhomogeneous terrain, with land and sea, mountain and plain, urban and forested areas, as shown in the terrain map of domain 2 (southern China) in Fig. 1b. The red dotted line in Fig. 1b represents the vertical cross-section domain that intercepts the most polluted part of the PRD and follows the prevailing (north-east) wind direction. The vertical domain is used to illustrate the vertical distribution of the $N_2O_5$ and $ClNO_2$ concentrations and the impacts of $N_2O_5$ uptake processes in southern China.

**2.3.2 Simulation cases**

Three simulation cases, shown in Table 2, were conducted from November 15 to December 5, 2013, during which $ClNO_2$ and $N_2O_5$ levels were measured at the TMS site. All simulations used ISORROPIA II as the aerosol thermodynamics module. Note that the Base case did not include $N_2O_5$ heterogeneous uptake (or $ClNO_2$ chemistry). The HET+Cl case included the complete $N_2O_5$ uptake and Cl activation processes, i.e., $N_2O_5$ loss on aerosol, $ClNO_2$ production and gaseous chlorine reactions. Differences in chemical concentrations between the Base and HET+Cl cases, i.e., HET+Cl-Base, represent the impacts of $N_2O_5$ uptake and Cl activation. To estimate the relative contribution of $N_2O_5$ uptake versus Cl activation to $O_3$ and $NO_y$ partitioning, the HET case was also conducted, which included $N_2O_5$ uptake but not $ClNO_2$ production (i.e. $ClNO_2$ yield, $\phi$, was set to 0) and therefore producing only nitrate from $N_2O_5$ uptake. The changes from the Base case to the HET case (HET-Base) represent the impacts of $N_2O_5$ heterogeneous uptake, whereas the changes from the HET case to the HET+Cl case (HET+Cl-HET), represent the impacts of Cl activation.

**3 Results and discussion**

**3.1 Model performance of WRF-Chem without $N_2O_5$ uptake and Cl activation**

The meteorological simulation determines the simulations of the transport of the air pollutants and therefore is crucial to the simulations of the spatial distributions of the atmospheric chemical species and their impacts. The performance of the meteorological module during the study period has been validated in Wang et al. (2016), which showed that atmospheric flow and other meteorological parameters were satisfactorily simulated. The reader is referred to Wang et al. (2016) for further details.

The chemical simulation results of WRF-Chem without $N_2O_5$ heterogeneous uptake and Cl activation, i.e., the Base case, were validated against hourly observations of several major air pollutants measured at 11 HKEPD stations and at the TMS site. $PM_{2.5}$, $NO_2$ and $O_3$ were selected as the validation species because they act as the reaction surface ($PM_{2.5}$) or precursors ($NO_2$



and $O_3$) for $N_2O_5$ and $ClNO_2$ production. As shown in Table 3, the validation results for HKEPD stations indicate that Base case simulated the major air pollutants reasonably well in this region but overestimated $PM_{2.5}$, slightly underestimated $NO_2$ and underestimated $O_3$. It should be noted that the technique for measuring $NO_2$ by the HKEPD, which is similar to that used in the regular air monitoring networks in North America and Europe, employs catalytic conversion which over-measures $NO_2$ (e.g., Xu et al., 2013). The discrepancy between the simulated and observed major air pollutants in this area is expected to affect the simulations of $N_2O_5$ and $ClNO_2$, which will be discussed in Section 3.2.1. In the Base case, the model, in general, satisfactorily reproduced the observed $PM_{2.5}$, $NO_2$ and $O_3$ levels at the TMS site during the nights when $N_2O_5$ and $ClNO_2$ were measured (Fig. S1). The capture of the temporal variations of these pollutants at the TMS site provides a good basis for simulation of the $N_2O_5$ and $ClNO_2$ temporal patterns (see Section 3.2.1). The model performance of major air pollutants of Base case is within the acceptable range and is similar to our previous applications of WRF-Chem (Zhang et al., 2015; Zhang et al., 2016) and other WRF-Chem model studies (e.g., Li et al., 2011).

The simulated fine chloride concentrations in the Base case were compared with observations from several campaigns, as shown in Table 4. Tan et al. (2009) reported average concentrations of 1.19 µg m⁻³ and 8.37 µg m⁻³ at an urban site in Guangzhou (GZ) in the PRD on normal days and hazy days in winter, respectively; in comparison, the Base case simulated an average level of 2.51 µg m⁻³ at that location. Tao et al. (2014) reported an average level of 3.30 µg m⁻³ in winter at the station of South China Institute of Environmental Science (SCIES) in Guangzhou; Base case predicted 2.13 µg m⁻³ at this location. At the Tung Chung (TC) site in Hong Kong, we had previously measured an average level of 1.10 µg m⁻³ of chloride in $PM_{2.5}$ in late autumn and early winter, while the Base case simulated 0.32 µg m⁻³. At the TMS site, an average level 0.37 µg m⁻³ of chloride was observed during the campaign (which was also the simulation period of this study), while the Base case predicted 0.14 µg m⁻³. The simulated chloride level in the Base case was in order with observations over the PRD, but it still under-simulated the observed chloride level due to the expected underestimates of chlorine sources in the EI we applied (see Section 2.1.2).

Overall, the validations of the meteorological and chemical simulations suggest that the model is capable of capturing the general characteristics of air flow and key atmospheric chemical processes and hence can provide a basis for further simulation of the distributions of the $N_2O_5$ and $ClNO_2$ concentrations, and the impacts of $N_2O_5$ uptake and Cl activation on $NO_y$ partitioning and $O_3$ production.

### 3.2 Simulation of $N_2O_5$ and $ClNO_2$ with $N_2O_5$ uptake and Cl activation

### 3.2.1 Comparison of simulated $N_2O_5$ and $ClNO_2$ with observation

The average observed and simulated (HET+Cl case) concentrations of $N_2O_5$ and $ClNO_2$ were calculated for each night, as shown in Fig. 2. The mean observed $N_2O_5$ concentrations for each night varied from 0.02 to 0.74 ppb during the study period, while the average simulated $N_2O_5$ values from the HET+Cl case were between 0.02 and 0.35 ppb. The HET+Cl case



reproduced the order of $N_2O_5$ concentrations but underestimated them within a factor of three. For $ClNO_2$, the average observed concentrations varied from 0.01 to 0.39 ppb, whilst the mean simulated values for each night varied between 0.05 and 0.42 ppb. The HET+Cl case reproduced the order of $ClNO_2$ concentrations with an overestimate mostly within a factor of four. The simulated and observed hourly concentrations of $N_2O_5$ and $ClNO_2$ are shown in Fig. S2, indicating that the HET+Cl case well captured the temporal variations of these two compounds.

The under-simulated $N_2O_5$ and over-simulated $ClNO_2$ values in the HET+Cl case point to the underestimation of the sources and/or the overestimation of the sink of $N_2O_5$ and the overestimation of the production of $ClNO_2$. As shown in section 3.1, the simulated $NO_2$ and $O_3$ levels in the HK-PRD region are lower than the observations, which results in lower production of $N_2O_5$; the simulated $PM_{2.5}$ concentrations are higher than the observed values which would lead to an overestimate of $N_2O_5$ heterogeneous loss. The observation-derived $N_2O_5$ uptake coefficients at the TMS site (Brown et al., 2016) varied from 0.004 to 0.029 with an average value of 0.014, whilst the simulated uptake coefficients ranged from 0.008 to 0.031 with an average of 0.019, which suggests that the HET+Cl simulation generally overestimates $N_2O_5$ uptake coefficients, which causes further overestimation of the loss of $N_2O_5$. The overestimated loss of $N_2O_5$ on aerosol inherently overestimated the production of $ClNO_2$. The parameterisations used in this study are likely to overestimate the $ClNO_2$ yield (Kim et al., 2014; Ryder et al., 2015), which would further overestimate the production of $ClNO_2$.

Discrepancies between the measured and simulated $N_2O_5$ and $ClNO_2$ levels have also been reported in previous model studies. Lowe et al. (2015) used the same parameterisations for $N_2O_5$ uptake that we applied in our study and showed slightly higher average simulated $N_2O_5$ values along two flight tracks but a factor of 1-2 lower simulated $N_2O_5$ in another flight. They noted that the underestimated $N_2O_5$ could be attributed to inaccuracies in the meteorological simulation. Sarwar et al. (2012) used the parameterisation for $N_2O_5$ uptake proposed by Davis et al. (2008) and by Bertram and Thornton et al. (2009) and yielded a slightly higher simulated peak value of $ClNO_2$ than the observed value in field studies conducted at different times from the model simulations. The authors attributed the overestimate of $ClNO_2$ to the overestimated $N_2O_5$ uptake in the parameterisations. Sarwar et al. (2014) predicted lower peak values of $ClNO_2$ than the observations and suggested that the underestimated $ClNO_2$ could be attributed to a relatively low model resolution (108 km).

### 3.2.2 Spatial distribution of average simulated $N_2O_5$ and $ClNO_2$

Figure 3a and 3c show the average mixing ratios of $N_2O_5$ and $ClNO_2$ during the entire simulation period within the lowest 1000 m (the approximate height of the PBL at noon) in southern China in the HET+Cl case. Elevated levels of $N_2O_5$ (>0.10 ppb) and $ClNO_2$ (>0.25 ppb) were predicted in the areas downwind of the PRD, as a result of the transport of pollutant enriched air masses towards the south-west of the PRD by the prevailing north-easterly winds. The areas with the highest simulated $N_2O_5$ and $ClNO_2$ values did not cover the TMS site at which the highest ever reported $N_2O_5$ and $ClNO_2$ values were observed

(Brown et al., 2016; Wang et al., 2016), which supports our speculation that the locations downwind of the PRD under the dominant north-easterly winds may frequently have higher levels of ClNO$_2$.

The vertical distributions of N$_2$O$_5$ and ClNO$_2$ in the vertical domain (as described in section 2.3.1) are shown in Fig. 3b and 3d. Elevated levels of N$_2$O$_5$ (> 0.10 ppb) were predicted up to around 1000 m a.g.l., with the highest N$_2$O$_5$ level (>0.25 ppb) mostly between 400-800m a.g.l., probably due to suppression of NO$_3$ (and N$_2$O$_5$) by NO in the lowest several hundred meters over the urban area, as shown in Fig. S3. Elevated levels of ClNO$_2$ (>0.25 ppb) were simulated up to 1000 m a.g.l., with the highest ClNO$_2$ values (>1.00 ppb) mostly concentrated within the near-surface layer of 0-200 m a.g.l. The vertical distribution of ClNO$_2$ was consistent with the vertical profile of chloride, as shown in Fig. S4. The simulated vertical distribution of N$_2$O$_5$ and ClNO$_2$ are similar to those of Sarwar et al. (2012), which showed that simulated N$_2$O$_5$ peaked at 200-400 m a.g.l., and simulated ClNO$_2$ peaked at the surface and stretched up to 400 m a.g.l. in several U.S. cities at dawn.

### 3.2.3 Dynamic evolutions in cases with typical and extreme meteorological conditions

We examine the time evolution of the spatial distribution of ClNO$_2$ in two cases. In the typical case (the night of December 1/2), southern China was dominated by consistent north-easterly winds which represented the average dynamic conditions during the study period, while in the extreme case (the night of December 3/4), the air-flow over the region abruptly changed. Note that in this extreme case, the highest ever-reported ClNO$_2$ levels were observed at the TMS site, and the back trajectories and observations of chemical species pointed to the transport to the site of well-processed plumes from the PRD with enriched anthropogenic chloride and other pollutants (Wang et al., 2016).

In the typical case, consistent north-easterly winds controlled southern China throughout the night (Fig. 4). At the beginning of the night (18:00, local time (LT)), ClNO$_2$ began to build up near the urban area (Fig. 4a) and near the surface (Fig. 4b); at midnight (00:00, LT), the air with an elevated level of ClNO$_2$ moved to coastal areas (Fig. 4c) and accumulated near the surface (Fig. 4d); at dawn (06:00, LT), the peak ClNO$_2$ level was predicted over the open sea (Fig. 4e), and pumped up to higher altitudes with the peak value near the surface (Fig. 4f), due to the higher boundary layer height over the ocean, as shown in Fig. S5.

In the extreme case, at the beginning of the evening (18:00, LT), southern China had unfavourable dispersion conditions over the land, including inconsistent wind directions and low wind speeds. The air pollutants emitted from the PRD slowly swirled over it, as shown in Fig. 5a, resulting in a longer 'cooking' time for ClNO$_2$ production. The vertical distribution (Fig. 5b) shows that ClNO$_2$-enriched air stretched from the ground up to 800m a.g.l. The enhanced production of ClNO$_2$ is believed to be partially responsible for the highest ClNO$_2$ mixing ratios measured at the TMS site at this night. At midnight (00:00, LT), inconsistent wind directions presented between land and sea areas: northerly winds dominated over the land area, while north-easterly winds dominated over the sea, leading to relatively slow motion of the ClNO$_2$-enriched plume from the land towards the ocean (Fig. 5c). The vertical distribution (Fig. 5d) suggests that ClNO$_2$ built up within the residue layer. At dawn (06:00,





LT), the north-easterly wind regained control over the land areas, and the air with the elevated level of $ClNO_2$ (>2.00 ppb) was driven towards the ocean, as shown in Fig. 5e. The vertical distribution (Fig. 5f) shows that the peak $ClNO_2$ concentration was predicted to be in the residue layer at ~300m a.g.l. The changes of wind flow over the region during this night resulted in abnormal changes in the history of the air masses that reached the TMS site and led to the abrupt changes in the air pollutants concentrations observed there (see Wang et al., 2016 for details).

From these results, it can be seen that the vertical distributions of $ClNO_2$ demonstrated distinct features in the two cases. To understand the underlying cause, it is of significance to measure the vertical profiles of $ClNO_2$ under various meteorological conditions. In addition, during the extreme event, the location with the highest predicted $ClNO_2$ (>2.00 ppb) was not at the TMS site (>1.00 ppb), but was located in the western parts of the PRD (i.e., the cites of Jiangmen and Zhaoqing), which supports the contention that the $ClNO_2$ concentrations at other locations could be even higher than those observed at the TMS site (Wang et al., 2016). It would be of great interest to conduct measurements at the areas where the highest $ClNO_2$ concentrations are predicted.

### 3.3 Impacts of $N_2O_5$ heterogeneous uptake and Cl activation on $NO_x$, total nitrate and $O_3$

### 3.3.1 Impacts in the horizontal and vertical domains

Figure 6 shows the simulation results for the average NO, $NO_2$, total nitrate and $O_3$ concentrations within the PBL (<1000m) in the Base case, and the difference of the results between the HET+Cl and Base cases in the horizontal domain. Elevated levels of NO (up to 26.18 ppb; Fig. 6a), $NO_2$ (up to 29.18 ppb; Fig. 6c), total nitrate (up to 23.43 µg m$^{-3}$; Fig. 6e), and $O_3$ (up to 44.50 ppb; Fig. 6g) were predicted in southern China in the Base case. With the prevailing north-easterly wind, the pollutants emitted from the PRD were transported towards the south-west, resulting in the most polluted regions being the PRD and its south-westerly downwind areas. After addition of the $N_2O_5$ uptake and Cl activation processes, the NO (Fig. 6b) and $NO_2$ (Fig. 6d) levels were significantly decreased in the entire domain by up to 1.93 ppb (~ 7.4%) and 4.73 ppb (~ 16.2%), respectively. The regions with greater impacts on the NO and $NO_2$ due to the added processes were mostly urban and suburban areas with large emissions of $NO_x$. A significant portion of $NO_x$ was transformed into total nitrate, which increased by as much as 13.45 µg m$^{-3}$ (~ 57.4%) through the heterogeneous uptake of $N_2O_5$ (see Fig. 6f). As can be seen in Fig. 6h, the $N_2O_5$ uptake and Cl activation noticeably increased $O_3$ levels across southern China, with a maximum increase up to 7.23 ppb (~16.3%). It is worth noting that in addition to the urban and suburban areas, the $O_3$ levels over the rural and coastal areas was also significantly affected by the added processes.

Figure 7 shows the average simulated NO, $NO_2$, total nitrate and $O_3$ values in the Base case, and the difference in the results between the Base and HET+Cl cases in the vertical domain. NO and $NO_2$ were concentrated within 800 m a.g.l. over the PRD and in downwind areas (see Fig. 7a and 7c). Total nitrate accumulated near the ground and stretched up to 800 m a.g.l. (Fig. 7e). Due to the titration effect of NO, relatively low average values of $O_3$ were simulated over urban areas (Fig. 7g). As shown





in Fig. 7b and 7d, the $N_2O_5$ uptake and Cl activation decreased the NO and $NO_2$ levels across the vertical domain, with the largest impacts seen in the near-surface layer (0-400 m a.g.l.) over the PRD. The lost NO and $NO_2$ were mostly transformed into total nitrate, which increased remarkably in the near-surface layer (Fig. 7f). The impacts of $N_2O_5$ uptake and Cl activation on the $O_3$ level varied with altitude: the $O_3$ increased throughout the lowest 800 m with the largest enhancement near the

ground, whereas it decreased above 1000 m a.g.l. (Fig. 7h). The changes in the $O_3$ were attributed to the combined effects of $NO_x$ loss due to $N_2O_5$ uptake and Cl atom production due to Cl activation, both of which have nonlinear impacts on $O_3$ production. The relative contribution of $N_2O_5$ uptake versus Cl activation on the NO, $NO_2$, total nitrate and $O_3$ concentrations will be discussed in the following section.

### 3.3.2 Relative contribution of $N_2O_5$ heterogeneous uptake versus Cl activation

To understand the relative contribution of $N_2O_5$ uptake and Cl activation, we conducted a sensitivity case (HET case as listed in Table 2) in which only nitrate was produced via $N_2O_5$ uptake. The differences in the simulations between the Base and HET cases represent the effects of $N_2O_5$ uptake, while those between the HET and HET+Cl cases represent the effects of Cl activation.

As shown in Fig. 8a and 8c, the mere consideration of nitrate production from $N_2O_5$ uptake led to decreases in NO and $NO_2$

by up to 1.11 ppb and 4.28 ppb, respectively. In addition, most of the lost $NO_x$ was transformed into total nitrate which increased by as much as 14.92 µg m$^{-3}$ (Fig. 8e). These results are similar to those of Lowe et al. (2015), who suggested that the nitrate in $PM_{10}$ was enhanced by up to 31.4% (increasing from 3.5 µg Kg$^{-1}$ to 4.6 µg kg$^{-1}$ at night) after considering the heterogeneous uptake processes of $N_2O_5$. The $N_2O_5$ uptake increased the $O_3$ levels by as much as 3.10 ppb in urban and suburban areas and decreased the $O_3$ by up to 1.47 ppb in rural and coastal areas (Fig. 8g), due to the nonlinearity of $O_3$

production, which are similar to the findings of Riemer et al. (2003) indicating that the $N_2O_5$ uptake resulted in an increase in the $O_3$ level in high-$NO_x$ areas and a decrease in low-$NO_x$ areas.

The further addition of Cl activation led to decreases in the NO level by as much as 0.96 ppb, as shown in Fig. 8b, and increases in $NO_2$ by up to 0.72 ppb, as shown in Fig. 8d. The Cl activation slightly decreased the total nitrate by up to 2.35 µg m$^{-3}$ (Fig. 8f), because a fraction of $N_2O_5$ was consumed to produce $ClNO_2$ in competition with nitrate production. The simulated $O_3$ was

significantly increased throughout the domain by as much as 4.54 ppb (Fig. 8h), which could be attributed to the effects of the activation of Cl radicals that initiated VOCs degradation and $O_3$ formation. The increase in the $O_3$ further enhanced the oxidation of NO into $NO_2$, and the recycling of $NO_2$ via $ClNO_2$ photolysis also contributed to the increase in $NO_2$.

### 3.4 Impacts of $N_2O_5$ uptake and Cl activation on $NO_y$ partitioning

The composition and partitioning of $NO_y$ affect the spatial range that nitrogenous species can reach after emission, and are

therefore of great importance in atmospheric chemistry (Bertram et al., 2013). The average concentration of $NO_y$ altered



modestly within the PBL over domain 2 with the addition of $N_2O_5$ uptake and Cl activation (12.24 ppb and 11.42 ppb in the Base and HET+Cl cases, respectively). The fractions of each species in $NO_y$, however, were substantially affected. The $NO_y$ partition was calculated over domain 2 for the Base and HET+Cl cases (see Fig 9). The percentage of $N_2O_5$ in $NO_y$ decreased from 7.80% in the Base case to 1.01% in the HET+Cl case, and that for $NO_3$ decreased from 0.38% to 0.09%. The $N_2O_5$ uptake and Cl activation reduced the fraction of NO from 9.59% to 6.84% and that of $NO_2$ from 51.07% to 35.17%. The percentage of total nitrate (nitrate + $HNO_3$) in $NO_y$ was significantly increased from 27.5% (=9.6%+17.9%) to 48.6% (=16.0%+32.6%). The added processes also introduced a new $NO_y$ species, $ClNO_2$, which accounted for 3.47% in the HET+Cl case. The decrease in the $NO_3$ level caused by $N_2O_5$ heterogeneous uptake would suppress the night-time chemistry of $NO_3$ and VOCs. The $N_2O_5$ uptake transferred a significant portion of $NO_x$ to total nitrate, reducing the lifetime and reaching range of $NO_x$-enriched plumes and thus affecting the $NO_x$-VOCs-$O_3$ photochemistry. The new species in the HET+Cl case, $ClNO_2$, contributed a non-negligible part of $NO_y$, and extended the lifetime and reaching range of reactive nitrogen.

**3.5 Sensitivity of $ClNO_2$ concentration to chlorine emission**

The production of $ClNO_2$ depends on the chloride concentration in aerosol according to the parameterisation used in this study (Bertram and Thornton, 2009). The only available chlorine EI for China is taken from a global dataset with a relatively low resolution (1°x1°) and for the year of 1990 (Keene et al., 1999). To test the sensitivity of the $ClNO_2$ production to Cl emissions, we conducted two simulations in which the RCEI emission were reduced by half (HET+Cl+0.5RCEI) and doubled (HET+Cl+2.0RCEI). The simulations show that the ambient chloride concentrations responded almost linearly to the applied chlorine emissions (data not shown). The simulated $ClNO_2$ has a similar temporal pattern in different Cl emissions (Fig. S6). The $ClNO_2$ concentrations have positive but not linear correlation to Cl emission changes. As shown in Fig. 10, halving the Cl emissions leads to a 31% reduction in the simulated $ClNO_2$ level, whereas doubling the Cl emissions results in an average 31% increase of $ClNO_2$. The results indicate that simulation of $ClNO_2$ production is sensitive to chlorine emission. Therefore, future studies are needed to develop an up-to-date anthropogenic chlorine EI in China to better model $ClNO_2$ production, and to quantify its impact on atmospheric chemistry and air quality.

**4 Summary and conclusions**

In this study, a state-of-the-art chemical transport model (WRF-Chem) was further developed by incorporation of a widely-used aerosol thermodynamics model (ISORROPIA II), parameterisation of heterogeneous uptake of $N_2O_5$ and $ClNO_2$ production, and gas-phase chlorine chemistry. The revised model was used to simulate the spatial distributions of $N_2O_5$ and $ClNO_2$ and the impacts on $O_3$ and $NO_y$ partitioning over the HK-PRD region where high levels of $N_2O_5$ and $ClNO_2$ had been observed. The revised model was able to capture the temporal patterns and the magnitudes of the observed $N_2O_5$ and $ClNO_2$ at a mountain-top site in Hong Kong, but tended to under-simulate $N_2O_5$ and over-simulate $ClNO_2$ because of the



underestimates of $N_2O_5$ sources and overestimates of $N_2O_5$ sink and $ClNO_2$ production. Model simulations show that under average meteorological conditions, high values of $N_2O_5$ and $ClNO_2$ are concentrated in the south-west region to the urban areas of the PRD and vertically peak within the layer of 400-800 m a.g.l. and 0-200 m a.g.l, respectively. At the night of December 3/4 when the highest ever-reported $ClNO_2$ (4.7 ppb) was observed, the model suggested that the high levels of

$ClNO_2$ were concentrated in the residue layer (~300m a.g.l.) above the study region. The model simulations suggested that the region downwind of the urban PRD may experience higher levels of $ClNO_2$ than that observed at the TMS site. $N_2O_5$ uptake and Cl activation significantly decreased the levels of NO and $NO_2$ by up to 1.93 ppb (~7.4%) and 4.73 ppb (~16.2%), respectively, but increased the total nitrate level by as much as 13.45 µg m$^{-3}$ (~ 57.4%) and the $O_3$ by up to 7.23 ppb (~16.3%) within the PBL. Our results demonstrate the significant impacts of $N_2O_5$ uptake and $ClNO_2$ production on $NO_x$ lifetime,

secondary nitrate production, and $O_3$ formation and underscore the necessity of considering these processes in air quality models. Our simulations of $ClNO_2$ levels over southern China are sensitive to chlorine emissions, which suggests the need to develop a more reliable emission inventory of chlorine for better quantification of the $N_2O_5$/$ClNO_2$ chemistry and their impacts over China.

**Acknowledgements:**

The authors would like to thank China National Meteorological Center and Hong Kong Observatory for providing the meteorological data and Hong Kong Environmental Protection Department for providing the routine air pollutants measurement data and the emission inventory in Hong Kong. This study is supported by a Hong Kong Polytechnic University PhD studentship, General Research Fund of Hong Kong Research Grants Council (PolyU 153026/14P), and Collaborative Research Fund of the Hong Kong Research Grants Council (C5022-14G). Both the data and source code of the revised model

used in this study are available from the corresponding author (cetwang@polyu.edu.hk) upon request.

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





**Table 1. Model configuration**

| WRF-Chem modules | Parameterization options | Reference |
|---|---|---|
| Land surface | Noah Land Surface Model | Chen and Dudhia (2001) |
| PBL scheme | YSU | Hong et al. (2006) |
| Microphysics | Purdue Lin Scheme | Lin et al. (1983) |
| Cumulus | Grell 3-D ensemble | Grell and Devenyi (2002) |
| Shortwave and longwave radiation | RRTMG | Iacono et al. (2008) |
| Gas chemistry | RACM_ESRL | Updated version based on Stockwell et al. (1997) |
| Aerosol | MADE/VBS | Ahmadov et al. (2012) |
| Photolysis | Madronich | Madronich (1987) |

**Table 2. Simulation cases**

| Cases | Aerosol thermodynamics module | $N_2O_5$ and $ClNO_2$ chemistry |
|---|---|---|
| Base | ISORROPIA II | None |
| HET | ISORROPIA II | $N_2O_5$ heterogeneous uptake, no $ClNO_2$ production |
| HET+Cl | ISORROPIA II | $N_2O_5$ heterogeneous uptake, $ClNO_2$ production and gas-phase Cl reactions |

**Table 3. Comparison of chemical simulation with observation for Base case**

| | $PM_{2.5}$ ($\mu g\ m^{-3}$) | $NO_2$ (ppb) | $O_3$ (ppb) |
|---|---|---|---|
| Observation Average | 37.43 | 33.67 | 28.29 |
| Simulation Average | 48.08 | 28.81 | 15.06 |
| Normalized Mean Bias | 28.5% | -14.4% | -46.8% |
| Fac2 | 0.69 | 0.71 | 0.46 |

**Table 4. Comparison of measured and simulated (Base) chloride**

| Location | Period | Average measured concentration ($\mu g\ m^{-3}$) | Average simulated concentration ($\mu g\ m^{-3}$) |
|---|---|---|---|
| GZ | 2007/12/31 to 2008/1/12 normal day | 1.19 [a] | 2.51 |
| GZ | 2007/12/31 to 2008/1/12 haze day | 8.37 [a] | 2.51 |
| SCIES, GZ | 2009-2010 winter | 3.30 [b] | 2.13 |
| TC, HK | 2011/10/25 to 2011/12/7 | 1.10 [c] | 0.32 |
| TMS, HK | 2013/11/15 to 2013/12/5 | 0.37 [d] | 0.14 |

a: Tan et al., 2009; b: Tao et al., 2014; c: unpublished data; d: Wang et al., 2016.




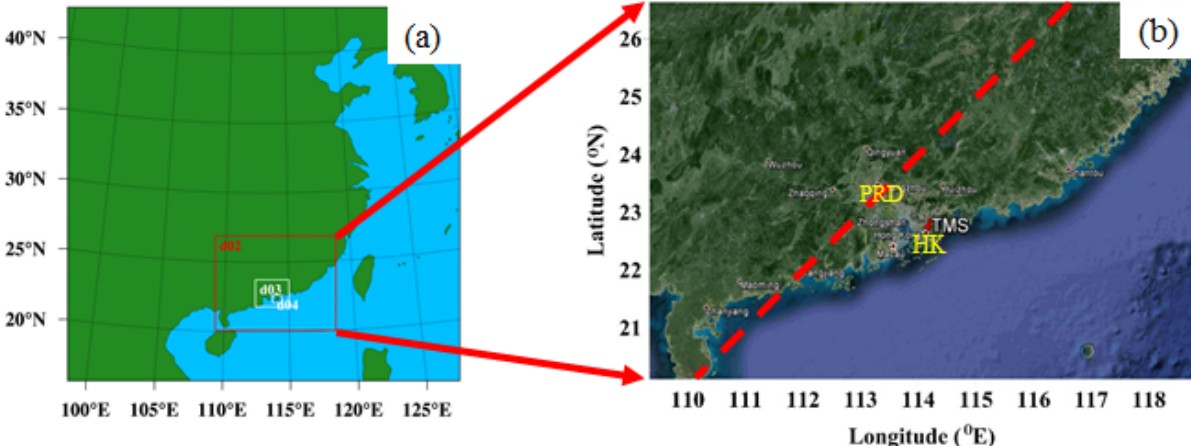

**Figure 1. (a) Domain settings of WRF-chem simulations, and (b) the terrain in domain 2 (southern China). The red dotted line represents the vertical domain that intercepts the most polluted PRD region along the prevailing wind direction (north-east). TMS is the location of the site where $N_2O_5$ and $ClNO_2$ were measured. HK and PRD are the general locations of Hong Kong and Pearl River Delta region.**





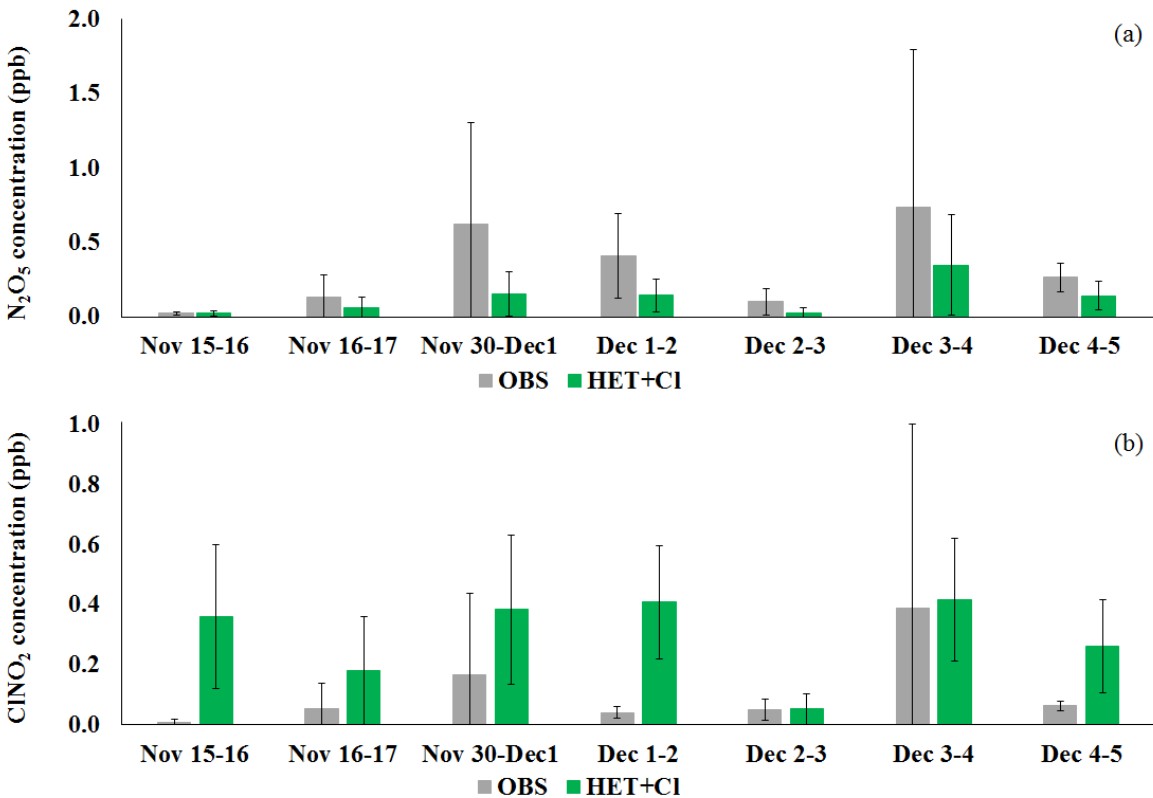

**Figure 2. Comparison of simulated and observed average (a) N$_2$O$_5$ and (b) ClNO$_2$ concentrations at each night at TMS site. Error bars represent the standard deviation.**



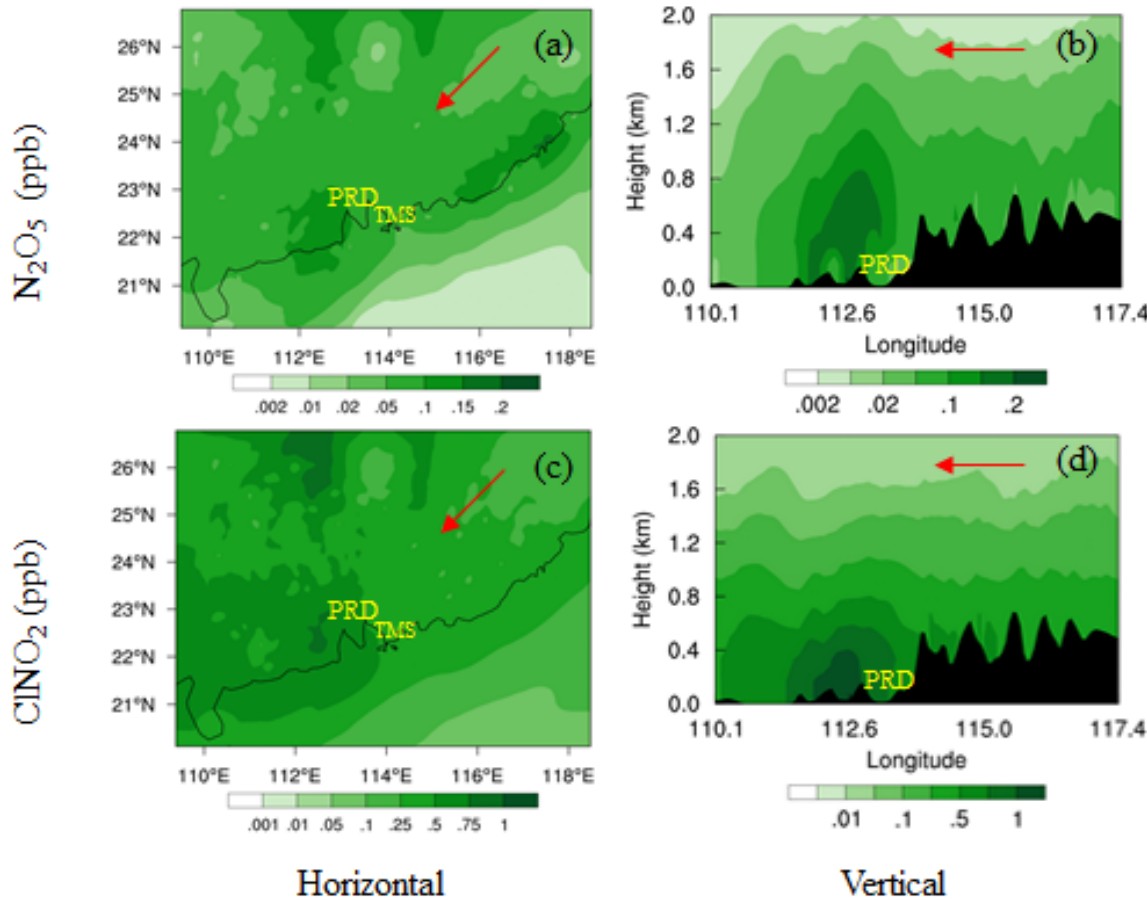

**Figure 3.** Horizontal distributions of (a) N$_2$O$_5$ and (c) ClNO$_2$ average mixing ratios (ppb) during the study period within the PBL from HET+Cl case; vertical distributions of (b) N$_2$O$_5$ and (d) ClNO$_2$ average mixing ratios (ppb) during the study period in the domain intercepting PRD and along the prevailing wind direction from HET+Cl case. Red arrows represent the prevailing wind direction.







**Figure 4. Horizontal distributions of ClNO₂ concentrations (ppb) at (a) 18:00 Dec 1, (c) 00:00 Dec 2, and (e) 06:00 Dec 2, LT within the PBL from HET+Cl case; vertical distributions of ClNO₂ concentrations (ppb) at (b) 18:00 Dec 1, (d) 00:00 Dec 2, and (f) 06:00 Dec 2, LT in the domain intercepting PRD and along the prevailing wind direction from HET+Cl case.**





**Figure 5. The same as in Figure 4, except at the night of December 3/4.**





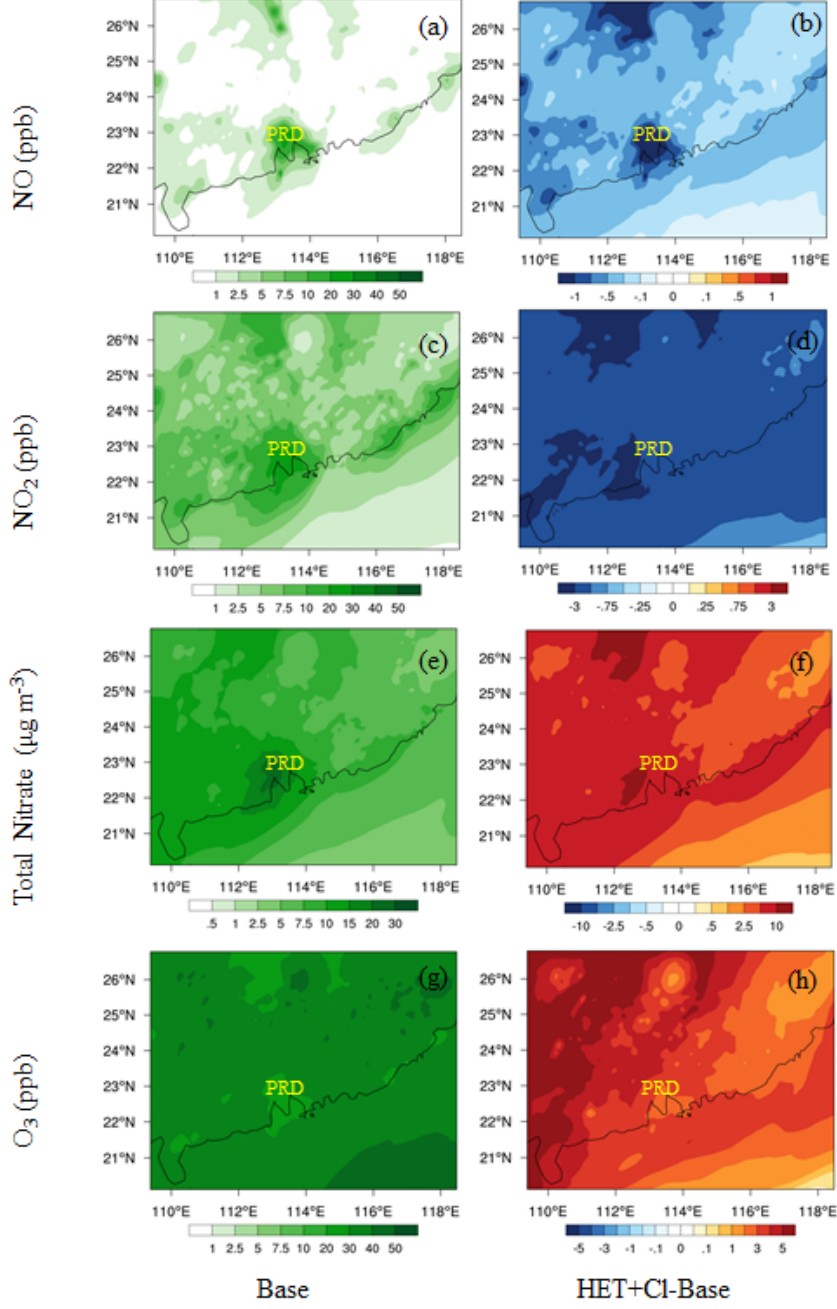

**Figure 6. Horizontal distributions of (a) NO (ppb), (c) NO₂ (ppb), (e) total nitrate (µg m⁻³) and (g) O₃ (ppb) average concentrations during the study period within the PBL from Base case; the average impacts of N₂O₅ uptake and Cl activation on (b) NO (ppb), (d) NO₂ (ppb), (f) total nitrate (µg m⁻³) and (h) O₃ (ppb) average concentrations during the simulation period in the horizontal domain within the PBL.**





**Figure 7. Vertical distributions of (a) NO (ppb), (c) NO₂ (ppb), (e) total nitrate (μg m⁻³) and (g) O₃ (ppb) average concentrations during the study period in the domain intercepting PRD and along the prevailing wind from Base case; the average impacts of N₂O₅ uptake and Cl activation on (b) NO (ppb), (d) NO₂ (ppb), (f) total nitrate (μg m⁻³) and (h) O₃ (ppb) average concentrations during the simulation period in the vertical domain.**



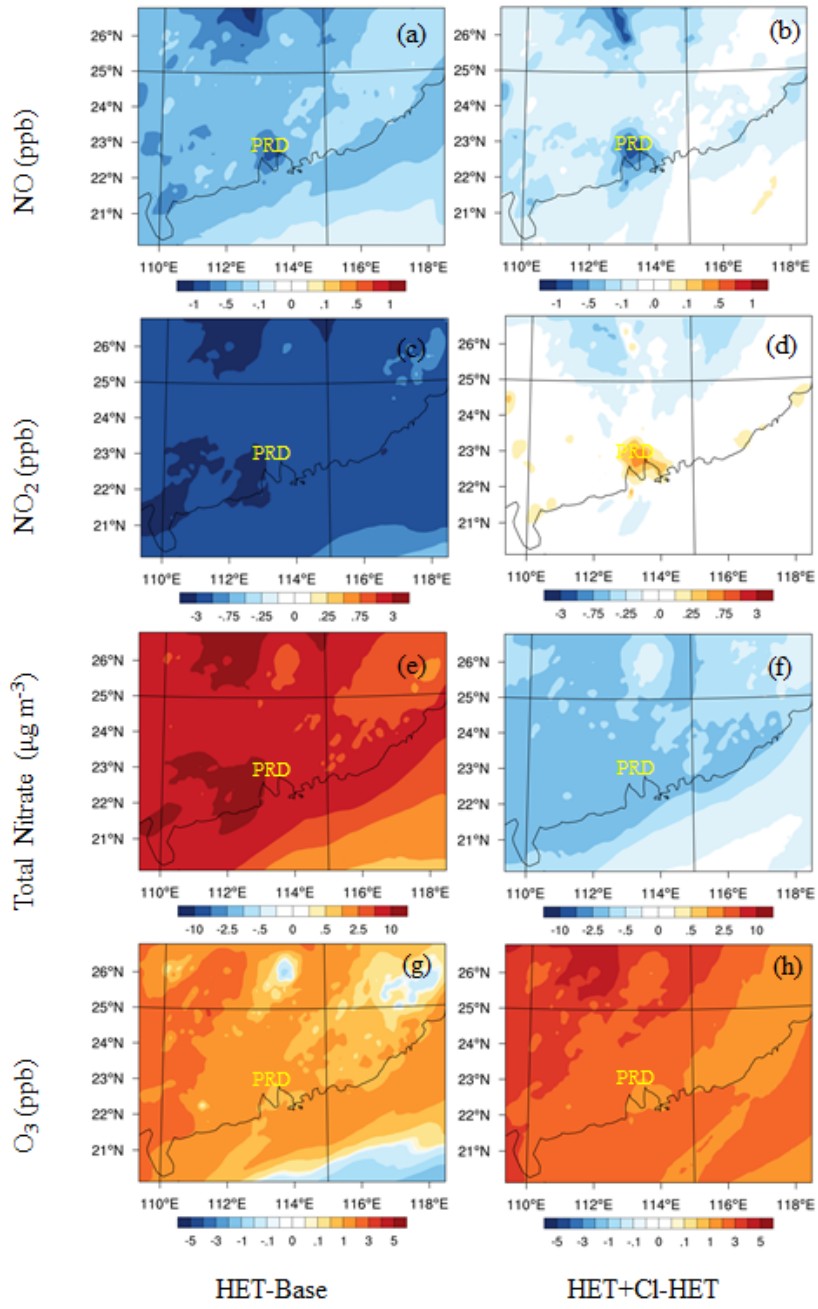

**Figure 8.** Average impacts of N$_2$O$_5$ heterogeneous uptake on (a) NO (ppb), (c) NO$_2$ (ppb), (e) total nitrate (μg m$^{-3}$) and (g) O$_3$ (ppb) average concentrations during the simulation period in the horizontal domain within the PBL; average impacts of Cl activation on (b) NO (ppb), (d) NO$_2$ (ppb), (f) total nitrate (μg m$^{-3}$) and (h) O$_3$ (ppb) average concentrations during the simulation period in the horizontal domain within the PBL.





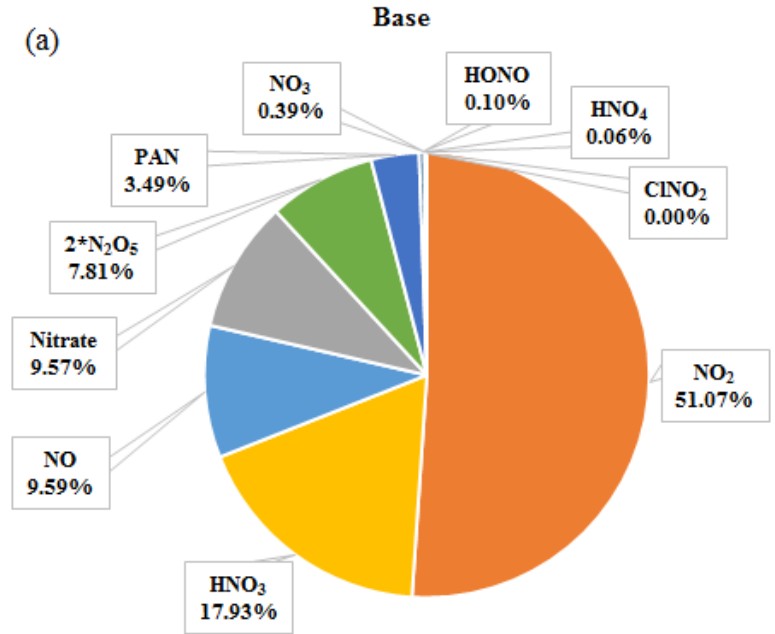

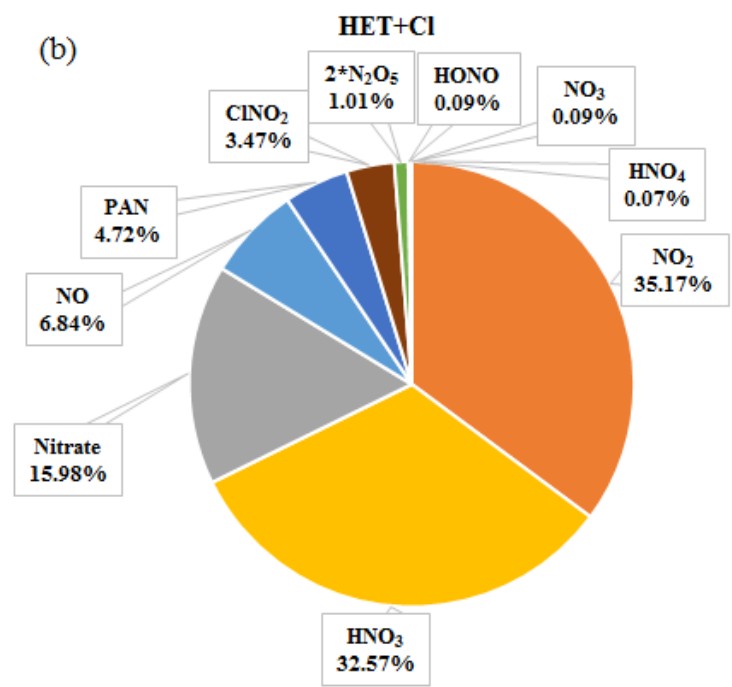

**Figure 9. Average NO$_y$ partitioning during the study period in southern China within the PBL as simulated in (a) Base and (b) HET+Cl case**





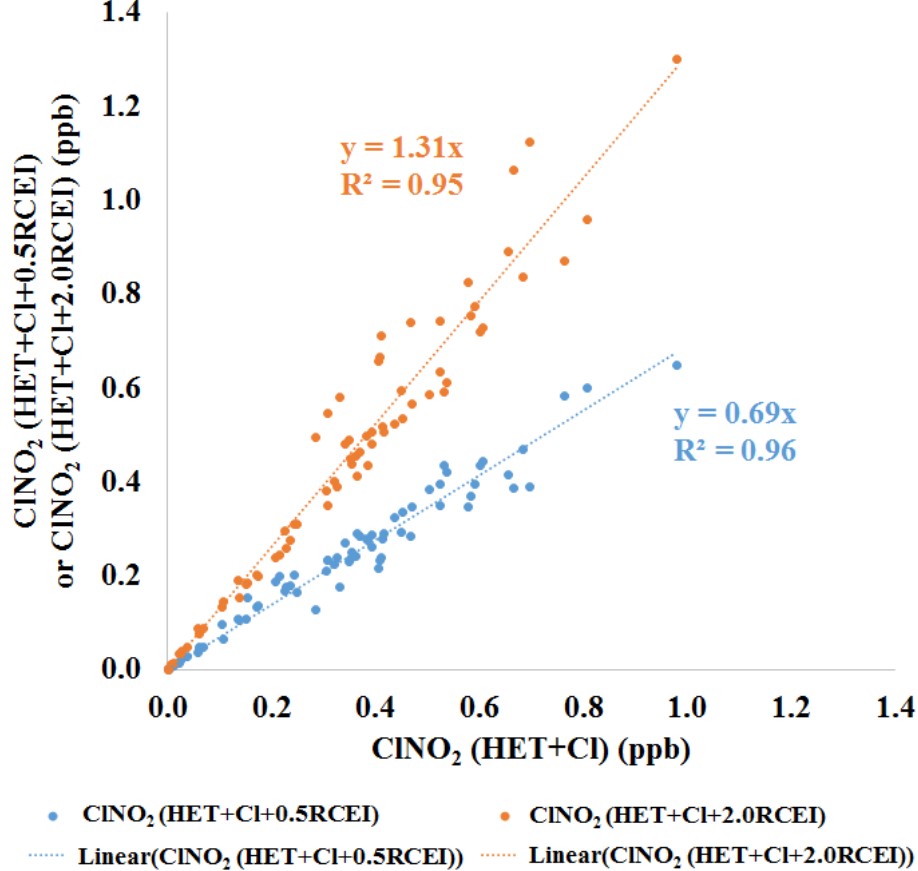

**Figure 10. Scatter plots of ClNO$_2$ (ppb) from simulations with half (HET+Cl+0.5RCEI) and twice (HET+Cl+2.0RCEI) RCEI emissions against ClNO$_2$ (ppb) from simulations with original RCEI emissions (HET+Cl).**

