# Peer review of "Impacts of heterogeneous uptake of dinitrogen pentoxide and chlorine activation on ozone and reactive nitrogen partitioning: Improvement and application of the WRF-Chem model in southern China"

_Atmospheric Chemistry and Physics, 2016_

## Referee Comment (RC1) · Anonymous Referee #1 · 27 Jun 2016

Li et al. present a set of regional photochemical modeling runs that simulate ClNO2 formation and impacts corresponding to recent field measurements in China. The field study reported the highest ever measured ambient concentration of ClNO2 indicating that this region of the world may be uniquely impacted by the chemistry associated with this compound. This study represents the first regional modeling study of ClNO2 impacts in Asia and is important for characterizing these impacts in a region with severe air pollution. The model is uniquely situated to provide a full spatial and temporal

characterization of this chemistry which is not feasible with measurements alone.

One major comment is that the model performance was not great and the authors often overstate the accuracy of the performance based on the comparisons provided. Rather than gloss over the poor model performance, the authors should acknowledge this limitation and discuss how those model inaccuracies might impact their results (for instance large under-estimates of PM2.5 might lead to underestimates of the ClNO2 formation). Despite the often poor model performance, this study is valuable since it is the first application of its kind in China and provides new insights into times and locations where ClNO2 impacts are predicted to be most important. This type of characterization of spatial and temporal patterns is not possible with measurements alone.

Another major comment is that all results (figs 3-10) are given as episode averages (all hours). Since many of the pollutants modeled have distinct diurnal profiles (i.e. O3, N2O5, ClNO2) these averages are hard to interpret. For N2O5 and ClNO2, the daytime values are essentially zero so these averages include high nighttime values averaged with many zeros during daytime hours. The reader does not get a good sense of the maximum magnitude of these pollutants at night. For ozone, many areas are titrated at night so again this doesn't give any sense of how high daytime ozone values are impacted. This averaging leads the authors to make statements like "elevated levels of . . . O3 (up to 44.5 ppb)" (p 11). 44.5 ppb of ozone is generally not considered an elevated level! I suggest that the authors add some results which either show diurnal averages of changes, time series of changes, or spatial plots of max values (and maximum changes) in addition to average values. This will provide a more complete picture of the modeled impacts of this chemistry.

Overall, this analysis used the best technical information currently available to complete this modeling and I think this paper will be of interest to ACP readers. I recommend publication after the authors address general comments above and specific comments below.

Specific comments:

Figs 3-7, and text throughout results section: Are heights given as above ground level (agl) or above sea level (asl)? The text repeatedly says "agl" but figures indicate terrain features in black which suggests that these heights are actually "asl". Please clarify and also add text to the caption which describes the black shaded regions in the figures.

Page 2, Line 16: hydrolysis of NO5 is "A" major loss pathway of NOx but perhaps not "THE" major loss pathway. What about reactions of NO3 with VOCs?

Page 2, Line 26: It would be good to clarify that gamma, the reactive uptake coefficient, represents the probability that a collision between N2O5 and a particle will result in uptake and chemical reaction.

Page 2, line 7 – Page 3, line 5: In the discussion of previous parameterizations for gamma, you should also mention that gamma has been measured by various field campaigns (Brown et al, 2006; Brown et al, 2009; Osthoff et al, 2008) which showed very different values for marine versus inland aerosols.

Page 3, lines 6-20: You missed several important earlier studies in your summary of modeled impacts of N2O5 and ClNO2 chemistry on air pollution concentrations/chemistry: Dentener and Crutzen, 1993; Riemer et al., 2003; Evans and Jacob, 2005; Simon et al, 2009

Page 3, line 20: change "biomass burning" to "biomass burning and sea salt"

Page 4, lines 18-24 and section 3.1: Comparisons with some additional measurements would be valuable if these measurements were made. For instance, if aerosol size distribution was measured that would allow for calculation of ambient surface area which could be directly compared with model results. Since surface area, not PM2.5 mass, drives this chemistry that would be a useful comparison. Also, were there any speciated PM2.5 measurements available to compare with the model (specifically aerosol nitrate and particulate chloride)? Were HCl and HNO3 measured? These could all

provide better constraints and characterization of model performance if they are available. In addition, a more complete model evaluation would be useful. For instance the authors might include timeseries of model performance, r2 values, maps of MB etc. There are several places in section 3.1 where the authors' characterization of the model performance is overly favorable and not supported by the figures provided. Statements that the model performed "reasonably well", "satisfactorily" etc are probably not warranted given that Figure S1 shows consistent under-predictions of ozone and PM2.5 of 20-40 ppb and 10-30 ug/m3 respectively.

Page 5, lines 5-15: Please specify if these are gas-phase or particle-phase chlorine emissions? Previous work on gas-phase chlorine emissions in the U.S. (Sarwar and Bhave, 2007; Chang et al, 2002) could be used as a starting point for deriving gas-phase chlorine emissions in China. Also, speciation profiles of PM2.5 emissions sources by Reff et al. (2009) could be used to derive particulate chloride emissions by applying fractional Cl contributions from all sources to the PM2.5 emissions in the current inventory. I am not suggesting that work needs to be done for this study, but the authors might discuss these past efforts as a basis for improving Chinese Cl emissions going forward.

Section 2.3.1: Please add information about which days were modeled. Was the modeled period Nov 15-Dec 5 to match measurements? Also, please state whether a spin-up period was included and, if so, how many days were used.

Page 9, lines 4-5: I don't think Fig S2 supports the contention that temporal variations of N2O5 and ClNO2 are "well captured". The model does predict that these pollutants build up at night and are close to 0 during daytime but other than that modeled peaks often appear at different hours and nights than observed peaks.

Page 9, lines 6-15: The reactive uptake coefficient could be too high in the model because the Bertram and Thornton parameterization does not account of organic inhibition of uptake that has been previously described by Riemer et al, 2009.

Page 12, line 13-14: The difference between HET and HET+Cl also shows the impact of lower levels of N2O5 conversion to HNO3.

Page 12, line 19: The decreases in O3 appear to only occur over a very small area, not over rural and coastal regions generally.

Page 12, line 22-23: This is confusing to me. If I understand the HET run correctly, it simply set the yield value to zero for the ClNO2 pathway which should mean that more N2O5 is converted to HNO3 and less is conserved in the ClNO2 reservoir. Therefore, the HET+CL simulation should increase NOx everywhere. What would cause broad decreases in NO and NO2 across the domain with the addition of the ClNO2 formation?

Page 12, line 23: Consider rephrasing, I don't consider a decrease of 2.35 ug/m3 "slight".

Page 12, line 25: What fraction of N2O5 produced ClNO2? It would be useful to report what yield values were predicted by the model for equation 5. How do these yields compare to previously reported observed yields (Osthoff et al., 2008) or modeled yields (Sarwar et al. 2012)? It might be useful to provide a map of yield values during nighttime hours.

Page 12, line 26-27: Simon et al. (2009) reported that half of the O3 impact from ClNO2 chemistry came from Cl activation while half came from the recycling of NO2.

Table 3: The authors should consider including O3 performance for daytime values (8-hr daily max or 1-hr daily max) as well as all hours averages. Also what is "fac2"? This is not defined anywhere in the paper. How is it calculated?

Table 4: The authors should state the time period used to calculated average simulated concentrations.

Figures 4 and 5: Consider using the same scale for the horizontal and vertical plots. The concentrations don't look different enough to warrant different scales.

Figure 6: The choice of the log scale makes variations in the O3 concentrations harder to see. Consider using a linear scale.

References:

Brown et al., 2006. "Variability in nocturnal nitrogen oxide processing and its role in regional air quality." Science, 411, 67-70.

Brown et al, 2009. "Reactive uptake coefficients for N2O5 determined from aircraft measurements during the Second Texas Air Quality Study: comparison to current model parameterizations." Journal of Geophysical Research – Atmospheres, 114, Article No. D00F10

Chang et al., 2002. "Sensitivity of urban ozone formation to chlorine emission estimates." Atmospheric Environment, 36, p. 4991-5003.

Dentener and Crutzen, 1993. "Reaction of N2O5 on tropospheric aerosols – impact on the global distributions of NOx, O3, and OH." Journal of Geophysical Research – Atmospheres, 98, D4, p. 7149-7163. Evans and Jacob, 2005. "Impact of new laboratory studies of N2O5 hydrolysis on global model budgets of tropospheric nitrogen oxides, ozone, and OH." Geophysical Research Letters, 23, 9, Article No L09813.

Osthoff et al, 2008. "High levels of nitryl chloride in the polluted subtropical marine boundary layer." Nature Geoscinces, 1 (5), 323-328.

Reff et al., 2009. "Emissions inventory of PM2.5 trace elements across the United States." Environmental Science & Technology, 43, 5790-5796.

Riemer et al, 2003. "Impact of the heterogeneous hydrolysis of N2O5 on chemistry and nitrate aerosol formation in the lower troposphere under photosmog conditions." Journal of Geophysical Research – Atmospheres, 108, D4, Article No 4144

Riemer et al., 2009. "Relative importance of organic coatings for the heterogeneous hydrolysis of N3O5 during summer in Europe." Journal of Geophysical Research –

[Figure]

Atmospheres, 114, Article No D17307

Sarwar and Bhave, 2007. "Modeling the effect of chlorine emissions on ozone levels over the eastern United States." Journal of applied meteorology and climatology, 46, 1009-1019.

Simon et al., 2009. "Modeling the impact of ClNO2 on ozone formation in the Houston area." Journal of Geophysical Research – Atmospheres, 114, Article No. D00F03.

---

## Referee Comment (RC2) · Anonymous Referee #2 · 20 Jul 2016

This paper presents WRF/Chem model simulations to assess the impacts of the heterogeneous hydrolysis of N2O5 on atmospheric chemistry for southern China, a region where high concentrations of N2O5 and ClNO2 were recently observed. A chlorine chemistry module was added to WRF/Chem to not only include HNO3 as a product of N2O5 hydrolysis, but also ClNO2, which in known to impact the oxidizing capability of the atmosphere by chlorine activation. The results show that for the chosen model domain and a simulation period during winter, N2O5 heterogeneous hydrolysis

contributes significantly to the formation of particulate nitrate and ozone. The results further point towards major model uncertainties due to chlorine emission inventories, which is consistent with previous studies.

The contribution of this work consists of WRF/Chem model development and the application of the extended model to a region where, so far, not much information on the importance of N2O5 hydrolysis has been available. Obtaining good agreement between simulation and observation of N2O5 and ClNO2 is challenging, so I commend the authors for their efforts. The study fits well within the scope of ACP, and it will be of interest for the community. I recommend the paper for publication after the authors address my questions and comments below.

1. page 2, line 25: $S_{\mathrm{aer}}$ is described as aerosol surface to volume ratio. This is confusing, it should rather be the aerosol surface area density, since it refers to the aerosol surface area per volume of air.

2. page 3, line 22: "1-minute value", what does that exactly mean? Were you sampling every minute or averaging over many samples for 1-minute intervals?

3. page 6, equation 2: The factor $A$ in this equation is a function of the surface area to volume ratio for the particles in those experiments. It would be worth checking that this is comparable to (or valid for) the study here.

4. How was the liquid water content of the aerosol determined? Are both inorganic and organic species contributing to aerosol water uptake, or is it only the inorganic species that determine the aerosol liquid water content?

5. Related to point 4, what is the liquid water content of the aerosols for the simulations presented here? Is the RH high enough that water uptake is predicted? For example, Lowe et al. (2015) and Chang et al. (2016) have shown that using the Bertram and Thornton parameterization can lead to problems in low RH environments — not because there is a problem with this parameterization, but rather with the way aerosol

water uptake is handled in CTMs. It would be interesting to see how this study compares in this regard.

6. Were clouds present during the simulation period and were they simulated? How is heterogeneous hydrolysis on cloud droplets handled?

7. page 7, line 3: Please add some information on the vertical model resolution. Many studies exist in the last 15 years that show pronounced gradient in N2O5 and NO3 mixing ratios, and the vertical resolution of the model is important. (e.g. Brown et al., 2007a, 2007b; Geyer and Stutz, 2004; Stutz et al., 2004, Riemer et al. 2003.)

8. Table 3: Explain "Fac2"

9. Table 3 and Figure S1: It sounds like the observations of PM2.5, NO2, and O3 are available for the entire period, not only for the nights when N2O5 and ClNO2 were observed. I suggest, for figure S1, to show the entire time series, which will convey better the information if the temporal variation of the pollutant is captured. With the gaps in the time series it's hard to tell.

10. What is the rationale for choosing the base case for the comparison to observations in section 3.1? This seems strange to me. I would assume that the HET+Cl case is the "best effort" to capture the processes that are occurring in the real atmosphere. So, what conclusion can be drawn from the comparison of observations to the base case? If the hydrolysis has an impact as the paper states, should we not expect a disagreement of base case and observations?

11. page 8, line 29: calculations of averages: which hours count as "night" for the presented case?

12. Figure 2: It would be interesting to add the "HET" case to this graph.

13. page 9, line 5, the statement: "the HET+Cl case captured the temporal evolution of the two compounds well". From Figure S2, I'm not sure if one can make such a statement. For some nights the peaks are roughly coinciding, for other nights not. I

realize that it is very difficult to obtain good agreement with these species. There can be many reasons why there are differences between a point measurement of ClNO2 and N2O5 and a model simulation, but I'd rather suggest not making such statements in a case like this.

14. To enhance the process-level analysis of this paper I suggest to comment on the spatial distribution of the yield $\phi$. Where in the model domain is it that ClNO2 is produced?

15. The terms "under-simulated" and "over-simulated" appear frequently in the manuscript. These are not the appropriate English terms. I suggest changing this to "underpredicted" and "overpredicted".

16. page 9, line 7: the overprediction of ClNO2 can also be due to an underestimation of the sinks.

17. General comments about the figures: They are very low resolution. I suggest to submit better-quality figures for the revised version.

18. page 9, line 26: "within the lowest 1000 m": Does this mean that the mixing ratios were averaged over the lowest 1000 m, or is one particular layer shown in Figure 3a and c? Please clarify.

19. page 9, line 10: Simulated uptake coefficients higher than observed ones: From the description in section 2.2.2 it appears that organic coatings are not taken into account even though it has been shown in several studies that the presence of these can lower the uptake coefficient notably. Could the presence of organics, which is not accounted for in the simulation, explain this discrepancy and consequently also the underprediction of N2O5 and overprediction of ClNO2? Please add some discussion.

19. page 10, line 5: change "suppression" to "reaction". NO3 also reacts with VOC. Does this also contribute to low NO3 concentrations near the ground?

20. page 10, line 9: reference to Sarwar et al (2012): Many studies have shown

evidence for pronounced vertical gradients in the profiles of N2O5 before that study, see my comment 7 above.

21. page 14, line 2: "average meteorological conditions": Remind the reader what this means (average in the sense of what?)

References Brown, S. S., W. P. Dube, H. D. Osthoff, D. E. Wolfe, W. M. Angevine, and A. R. Ravishankara (2007a), High resolution vertical distributions of NO3 and N2O5 through the nocturnal boundary layer, Atmos. Chem. Phys., 7, 139–149.

Brown, S. S., et al. (2007b), Vertical profiles in NO3 and N2O5 measured from an aircraft: Results from the NOAA P-3 and surface platforms during the New England Air Quality Study 2004, J. Geophys. Res., 112, D22304, doi:10.1029/2007JD008883.

Chang, W. L., S. S. Brown, J. Stutz, A. M. Middlebrook, R. Bahreini, N. L. Wagner, W. P. Dube, I. Pollack, T. B. Ryerson, and N. Riemer (2016), Evaluating N2O5 heterogeneous hydrolysis parameterizations for CalNex 2010, J. Geophys. Res. Atmos., 121, doi:10.1002/2015JD024737.

Geyer, A., and J. Stutz (2004), Vertical profiles of NO3, N2O5, O3, and NOx in the nocturnal boundary layer: 2. Model studies on the altitude dependence of composition and chemistry, J. Geophys. Res., 109, D12307, doi:10.1029/2003JD004211.

Lowe, D., Archer-Nicholls, S., Morgan, W., Allan, J., Utembe, S., Ouyang, B., Aruffo, E., et al. (2015). WRF-Chem model predictions of the regional impacts of N2O5 heterogeneous processes on night-time chemistry over north-western Europe. Atmospheric Chemistry and Physics, 15 1385-1409.

Riemer, N., H. Vogel, B. Vogel, B. Schell, I. Ackermann, C. Kessler, and H. Hass (2003), Impact of the heterogeneous hydrolysis of N2O5 on chemistry and nitrate aerosol formation in the lower troposphere under photosmog conditions, J. Geophys. Res., 108(D4), 4144, doi:10.1029/ 2002JD002436.

Stutz, J., B. Alicke, R. Ackermann, A. Geyer, A. White, and E. Williams (2004), Vertical profiles of NO3, N2O5, O3, and NOX in the nocturnal boundary layer: 1. Observations during the Texas Air Quality Study 2000, J. Geophys. Res., 109, D12306, doi:10.1029/2003JD004209.

---

## Author Comment (AC1) · 19 Aug 2016

**Response to Anonymous Referee #1**

We thank the referee for the comments and suggestions which help us improve the quality of the paper. Our response and the corresponding changes are listed below.

**General Comments:**

Li et al. present a set of regional photochemical modeling runs that simulate ClNO2 formation and impacts corresponding to recent field measurements in China. The field study reported the highest ever measured ambient concentration of ClNO2 indicating that this region of the world may be uniquely impacted by the chemistry associated with this compound. This study represents the first regional modeling study of ClNO2 impacts in Asia and is important for characterizing these impacts in a region with severe air pollution. The model is uniquely situated to provide a full spatial and temporal characterization of this chemistry which is not feasible with measurements alone.

One major comment is that the model performance was not great and the authors often overstate the accuracy of the performance based on the comparisons provided. Rather than gloss over the poor model performance, the authors should acknowledge this limitation and discuss how those model inaccuracies might impact their results (for instance large under-estimates of PM2.5 might lead to underestimates of the ClNO2 formation). Despite the often poor model performance, this study is valuable since it is the first application of its kind in China and provides new insights into times and locations where ClNO2 impacts are predicted to be most important. This type of characterization of spatial and temporal patterns is not possible with measurements alone.

Another major comment is that all results (figs 3-10) are given as episode averages (all hours). Since many of the pollutants modeled have distinct diurnal profiles (i.e. O3, N2O5, ClNO2) these averages are hard to interpret. For N2O5 and ClNO2, the daytime values are essentially zero so these averages include high nighttime values averaged with many zeros during daytime hours. The reader does not get a good sense of the maximum magnitude of these pollutants at night. For ozone, many areas are titrated at night so again this doesn't give any sense of how high daytime ozone values are impacted. This averaging leads the authors to make statements like "elevated levels of . . . O3 (up to 44.5 ppb)" (p 11). 44.5 ppb of ozone is generally not considered an elevated level! I suggest that the authors add some results which either show diurnal averages of changes, time series of changes, or spatial plots of max values (and maximum changes) in addition to average values. This will provide a more complete picture of the modeled impacts of this chemistry.

Overall, this analysis used the best technical information currently available to complete this modeling and I think this paper will be of interest to ACP readers. I recommend publication after the authors address general comments above and specific comments below.

**Response**: We agree with the reviewer that the model performance was not as great as we stated. We have revised the sentence describing model performance in the abstract to be 'The updated model can generally capture the temporal variation of $N_2O_5$ and $ClNO_2$ observed at a mountain-top site in Hong Kong, but overestimates $N_2O_5$ uptake and $ClNO_2$ production.' We also deleted the word 'satisfactorily' in section 3.1 (p8), and deleted a sentence in section 3.1 (p8) 'The capture

of the temporal variations of these pollutants at the TMS site provides a good basis for simulation of the $N_2O_5$ and $ClNO_2$ temporal patterns (see Section 3.2.1).'

We had discussed the potential effects of the discrepancy between the simulated and observed concentrations of $PM_{2.5}$, $NO_2$ and $O_3$ in Hong Kong – Pearl River Delta region on the $N_2O_5$ and $ClNO_2$ simulation in section 3.2.1.

We have replaced the Figure 3 with the average $ClNO_2$ and $N_2O_5$ concentrations during nighttime (18:00-07:00). We have added the spatial plots of average maximum values and maximum changes of NO, $NO_2$, total nitrate and $O_3$ in the supplement to provide extra information on the impacts of the $N_2O_5$ and $ClNO_2$ chemistry in southern China.

**Specific comments:**

**1.** Figs 3-7, and text throughout results section: Are heights given as above ground level (agl) or above sea level (asl)? The text repeatedly says "agl" but figures indicate terrain features in black which suggests that these heights are actually "asl". Please clarify and also add text to the caption which describes the black shaded regions in the figures.

**Response**: The statement used in text, 'agl', is correct. We calculated the 'agl' based on the 'asl' and the 'terrain height' (black shaded features in cross-section figures). We have added the description of the terrain features in figures.

**2.** Page 2, Line 16: hydrolysis of NO5 is "A" major loss pathway of NOx but perhaps not "THE" major loss pathway. What about reactions of NO3 with VOCs?

**Response**: Indeed, the loss of NOx through the $N_2O_5$ heterogeneous reaction and through the reaction between $NO_3$ with VOCs can be both important. The word 'the' has been changed to 'a'.

**3.** Page 2, Line 26: It would be good to clarify that gamma, the reactive uptake coefficient, represents the probability that a collision between N2O5 and a particle will result in uptake and chemical reaction.

**Response**: The following explanation of the reactive uptake coefficient has been added to the manuscript 'the possibility that a colliding of $N_2O_5$ molecule with a particle will lead to uptake and chemical reaction (Sarwar, et al. 2012).' in section 1 (p2).

**4.** Page 2, line 7 – Page 3, line 5: In the discussion of previous parameterizations for gamma, you should also mention that gamma has been measured by various field campaigns (Brown et al, 2006; Brown et al, 2009; Osthoff et al, 2008) which showed very different values for marine versus inland aerosols.

**Response**: Previous measurement studies on uptake coefficients have been added to the manuscript.

**5.** Page 3, lines 6-20: You missed several important earlier studies in your summary of modeled impacts of N2O5 and ClNO2 chemistry on air pollution concentrations/ chemistry: Dentener and Crutzen, 1993; Riemer et al., 2003; Evans and Jacob, 2005; Simon et al, 2009

**Response**: These previous studies of the impacts of N2O5 and ClNO2, including Dentener and Crutzen, 1993; Riemer et al., 2003; Evans and Jacob, 2005; Simon et al, 2009, have been added to the manuscript as suggested.

**6.** Page 3, line 20: change "biomass burning" to "biomass burning and sea salt"

**Response**: Corrected.

**7.** Page 4, lines 18-24 and section 3.1: Comparisons with some additional measurements would be valuable if these measurements were made. For instance, if aerosol size distribution was measured that would allow for calculation of ambient surface area which could be directly compared with model results. Since surface area, not PM2.5 mass, drives this chemistry that would be a useful comparison. Also, were there any speciated PM2.5 measurements available to compare with the model (specifically aerosol nitrate and particulate chloride)? Were HCl and HNO3 measured? These could all provide better constraints and characterization of model performance if they are available. In addition, a more complete model evaluation would be useful. For instance, the authors might include time series of model performance, r2 values, maps of MB etc. There are several places in section 3.1 where the authors' characterization of the model performance is overly favorable and not supported by the figures provided. Statements that the model performed "reasonably well", "satisfactorily" etc are probably not warranted given that Figure S1 shows consistent under-predictions of ozone and PM2.5 of 20-40 ppb and 10-30 ug/m3 respectively.

**Response**: The comparison of measured and simulated surface area at TMS site has been added in section 3.1. The comparison of aerosol nitrate at TMS site has been added in section 3.1. The comparison of observed and simulated chloride had been conducted in section 3.1. No gas-phase HCl and HNO$_3$ were measured.

Time series of the comparison of measured and simulated PM$_{2.5}$, NO$_2$, and O$_3$ at the environmental monitoring stations and at TMS site have been added to the supplement.

We have revised the description of the model performance, see our response to the general comment.

**8.** Page 5, lines 5-15: Please specify if these are gas-phase or particle-phase chlorine emissions? Previous work on gas-phase chlorine emissions in the U.S. (Sarwar and Bhave, 2007; Chang et al, 2002) could be used as a starting point for deriving gas-phase chlorine emissions in China. Also, speciation profiles of PM2.5 emissions sources by Reff et al. (2009) could be used to derive particulate chloride emissions by applying fractional Cl contributions from all sources to the PM2.5 emissions in the current inventory. I am not suggesting that work needs to be done for this study, but the authors might discuss these past efforts as a basis for improving Chinese Cl emissions going forward.

**Response**: These are both gas and particle phase. For biomass burning, it is aerosol phase chlorine emission. For anthropogenic emission, it is gas phase chlorine (HCl) emission.

We have added a short description on the methodology that could be used to develop the chloride emission inventory in China.

**9.** Section 2.3.1: Please add information about which days were modeled. Was the modeled period Nov 15-Dec 5 to match measurements? Also, please state whether a spin-up period was included and, if so, how many days were used.

**Response**: The simulation period was November 15 to December 5, 2013, which had been stated in Section 2.3.2 (p7). The reviewer's understanding is correct. The simulation period was chosen according to the measurement period. One-day spin-up period was used.

**10.** Page 9, lines 4-5: I don't think Fig S2 supports the contention that temporal variations of N2O5 and ClNO2 are "well captured". The model does predict that these pollutants build up at night and are close to 0 during daytime but other than that modeled peaks often appear at different hours and nights than observed peaks.

**Response**: We agree with the reviewer that the temporal variation of $N_2O_5$ and $ClNO_2$ was not well captured in our simulation. The words 'well' in 'well captured' has been revised to 'generally' in section 3.2.1 (p9).

**11.** Page 9, lines 6-15: The reactive uptake coefficient could be too high in the model because the Bertram and Thornton parameterization does not account of organic inhibition of uptake that has been previously described by Riemer et al, 2009.

**Response**: We agree the reviewer's suggestion. In fact, in page 9, we had stated that in Sarwar et al. (2012), the authors attributed the parameterization (Bertram and Thornton, 2009) to be a possible reason that $ClNO_2$ was overestimated.

The following sentence has been added to the manuscript "The reactive uptake coefficient could be overestimated because the parameterization used in this study (Bertram and Thornton, 2009) does not consider the inhibition of organic coating to the uptake coefficient." in section 3.2.1 (p9).

**12.** Page 12, line 13-14: The difference between HET and HET+Cl also shows the impact of lower levels of N2O5 conversion to HNO3.

**Response**: Indeed, the difference of HET and HET+Cl, i.e. the impacts of $ClNO_2$ production, showed that less $N_2O_5$ were transformed into $HNO_3$ (and nitrate aerosol). And in the manuscript, we used total nitrate ($HNO_3$+nitrate aerosol) to avoid redundant description.

**13.** Page 12, line 19: The decreases in O3 appear to only occur over a very small area, not over rural and coastal regions generally.

**Response**: We agree. The sentence has been modified accordingly.

**14.** Page 12, line 22-23: This is confusing to me. If I understand the HET run correctly, it simply set the yield value to zero for the ClNO2 pathway which should mean that more N2O5 is converted to HNO3 and less is conserved in the ClNO2 reservoir. Therefore, the HET+CL simulation should increase NOx everywhere. What would cause broad decreases in NO and NO2 across the domain with the addition of the ClNO2 formation?

**Response**: The reviewer's understanding is correct. In HET case, the $ClNO_2$ yield was set to be zero, and all $N_2O_5$ loss was transformed into total nitrate.

The possible causes for the changes of NO and $NO_2$ from HET to HET+Cl case are discussed below. The produced $ClNO_2$ *(1) releases $NO_2$* and Cl radical after sunrise, and both of them increase the formation of $O_3$. The increased $O_3$ enhances the reaction of NO with $O_3$, which leads to the *(2) loss of NO*, and the *(3) formation of $NO_2$.* The increased $O_3$ also increases the level of OH radicals, which enhances the reaction of $NO_2$ with OH and then results in the *(4) loss of $NO_2$*. The combination of the effects *(1), (2), (3) and (4)* mentioned above are the net effects of the $ClNO_2$ production on NO and $NO_2$. Our study showed that the NO was reduced across the domain. And for the $NO_2$, the enhanced production outweighed the loss in urban areas, while in other regions, the $NO_2$ was decreased.

We have added a short discussion of the possible causes in the manuscript.

**15.** Page 12, line 23: Consider rephrasing, I don't consider a decrease of 2.35 ug/m3 "slight".

**Response**: The word 'slight' has been deleted.

**16.** Page 12, line 25: What fraction of N2O5 produced ClNO2? It would be useful to report what yield values were predicted by the model for equation 5. How do these yields compare to previously reported observed yields (Osthoff et al., 2008) or modeled yields (Sarwar et al. 2012)? It might be useful to provide a map of yield values during nighttime hours.

**Response**: The spatial distribution of simulated yield during nighttime has been added to the manuscript. And the comparison of the yields with previous reported observed and simulated has been added to the manuscript. The following sentence has been added to the manuscript.

"The simulated yield of $ClNO_2$ during night-time within PBL ranged within 0.1-0.7, which is consistent with previous observation study (0.1-0.65) (Osthoff et al., 2008) and modelling study (0-0.9) (Sarwar et al., 2012)."

**17.** Page 12, line 26-27: Simon et al. (2009) reported that half of the O3 impact from ClNO2 chemistry came from Cl activation while half came from the recycling of NO2.

**Response**: The conclusion from Simon et al. (2009) has been added to explain our results.

**18.** Table 3: The authors should consider including O3 performance for daytime values (8-hr daily max or 1-hr daily max) as well as all hours averages. Also what is "fac2"? This is not defined anywhere in the paper. How is it calculated?

**Response**: We have added 1-h daily maximum $O_3$ as the indicator of model performance.

FAC2 is defined as the fraction of simulated results that are within a factor of two of the observations. The calculation of FAC2 is as follows.

FAC2 = (the number of simulations that are within a factor of two of the observations) / (the number of observations).

The definition of FAC2 has been added to the manuscript.

**19.** Table 4: The authors should state the time period used to calculated average simulated concentrations.

**Response**: The time period used to calculate the average simulated concentration is the entire simulation period (November 15 to December 5, 2013). It has been added to the manuscript.

**20.** Figures 4 and 5: Consider using the same scale for the horizontal and vertical plots. The concentrations don't look different enough to warrant different scales.

**Response**: The scales have been revised as suggested.

**21.** Figure 6: The choice of the log scale makes variations in the O3 concentrations harder to see. Consider using a linear scale.

**Response**: A linear scale has been applied to the $O_3$ concentration figure.

**Reference:**

Dentener, F.J. and Crutzen, P.J.: Reaction of $N_2O_5$ on tropospheric aerosols: Impact on the global distributions of $NO_x$, $O_3$, and OH, J. Geophys. Res. Atmos., 98, 7149-7163, 1993.

Evans, M.J. and Jacob, D.J.: Impact of new laboratory studies of $N_2O_5$ hydrolysis on global model budgets of tropospheric nitrogen oxides, ozone, and OH, Geophys. Res. Lett., 32, doi: 10.1029/2005GL022469, 2005.

Riemer, N., Vogel, H., Vogel, B., Schell, B., Ackermann, I., Kessler, C. and Hass, H.: Impact of the heterogeneous hydrolysis of $N_2O_5$ on chemistry and nitrate aerosol formation in the lower troposphere under photosmog conditions, J. Geophys. Res. Atmos., 108, doi: 10.1029/2002JD002436, 2003.

Sarwar, G., Simon, H., Bhave, P. and Yarwood, G.: Examining the impact of heterogeneous nitryl chloride production on air quality across the United States, Atmos. Chem. Phys., 12, 6455-6473, 2012.

Simon, H., Y. Kimura, G. McGaughey, D. T. Allen, S. S. Brown, H. D. Osthoff, J. M. Roberts, D. Byun, and D. Lee: Modeling the impact of $ClNO_2$ on ozone formation in the Houston area, J. Geophys. Res., 114, doi:10.1029/2008JD010732, 2009.

---

## Author Comment (AC2) · 19 Aug 2016

**Response to Anonymous Referee #2**

We would like to thank the referee #2 for the comments and suggestions which help us improve the manuscript. Our response and the corresponding changes are listed below.

**General Comments:**

This paper presents WRF/Chem model simulations to assess the impacts of the heterogeneous hydrolysis of N2O5 on atmospheric chemistry for southern China, a region where high concentrations of N2O5 and ClNO2 were recently observed. A chlorine chemistry module was added to WRF/Chem to not only include HNO3 as a product of N2O5 hydrolysis, but also ClNO2, which in known to impact the oxidizing capability of the atmosphere by chlorine activation. The results show that for the chosen model domain and a simulation period during winter, N2O5 heterogeneous hydrolysis contributes significantly to the formation of particulate nitrate and ozone.

The results further point towards major model uncertainties due to chlorine emission inventories, which is consistent with previous studies. The contribution of this work consists of WRF/Chem model development and the application of the extended model to a region where, so far, not much information on the importance of N2O5 hydrolysis has been available. Obtaining good agreement between simulation and observation of N2O5 and ClNO2 is challenging, so I commend the authors for their efforts. The study fits well within the scope of ACP, and it will be of interest for the community. I recommend the paper for publication after the authors address my questions and comments below.

**Specific comments:**

**1.** page 2, line 25: Saer is described as aerosol surface to volume ratio. This is confusing, it should rather be the aerosol surface area density, since it refers to the aerosol surface area per volume of air.

**Response:** The definition of Saer has been revised to 'aerosol surface area density'.

**2.** page 3, line 22: "1-minute value", what does that exactly mean? Were you sampling every minute or averaging over many samples for 1-minute intervals?

**Response:** '1-minute value' is the average of data points collected every 6 second in a 1-minute interval. Please refer to Wang et al. (2016) for further details on the CIMS measurement.

**3.** page 6, equation 2: The factor A in this equation is a function of the surface area to volume ratio for the particles in those experiments. It would be worth checking that this is comparable to (or valid for) the study here.

**Response:** The factor 'A' in the parameterization proposed by Bertram and Thornton (2009) refers to a pre-factor which includes the ratio of the *volume to surface area* of the particles used in their experiments which is $3.75*10^{-8}$ m, i.e. the ratio of the *surface area to volume* is $2.67*10^7$ m$^{-1}$.

The average simulated ratio of surface area to volume for the particles in southern China within the PBL is shown below (the interval is $0.5*10^7$ m$^{-1}$), from which we can see that the simulated ratio is practically within the range of 0.5 to 2.5 $*10^7$ m$^{-1}$ in southern China, which is very close to the value used in the parameterization.

We have added the comparison of observed and simulated ratio of the surface area to volume to the manuscript.

[Figure]

Figure R1. Average simulated ratio of surface area to volume (m$^{-1}$) for the particles within PBL

**4.** How was the liquid water content of the aerosol determined? Are both inorganic and organic species contributing to aerosol water uptake, or is it only the inorganic species that determine the aerosol liquid water content?

**Response:** The liquid water content was predicted using the thermodynamic module, ISORROPIA. Only the inorganic species have been considered in determining the aerosol liquid water content.

**5.** Related to point 4, what is the liquid water content of the aerosols for the simulations presented here? Is the RH high enough that water uptake is predicted? For example, Lowe et al. (2015) and Chang et al. (2016) have shown that using the Bertram and Thornton parameterization can lead to problems in low RH environments — not because there is a problem with this parameterization, but rather with the way aerosol water uptake is handled in CTMs. It would be interesting to see how this study compares in this regard.

**Response:** The spatial plot of aerosol liquid water content is shown below.

[Figure]

Figure R2. The average aerosol liquid water content ($\mu g\ m^{-3}$) in southern China during the simulation period within the PBL.

We had validated RH simulation performance, and the RH was well predicted during the simulation period (Wang et al., 2016). The mean bias between simulated and observed RH was only 3.54, and the correlation coefficient was 0.89, and the root mean square error was only 11.29.

We have noticed the conclusions of Lowe et al. (2015) and Chang et al. (2016), but it seems that our model performed relatively well to predict the uptake of $N_2O_5$ on aerosol surface in the Hong Kong – Pearl River Delta region, given that the simulated uptake at TMS (in the range of 0.008 to 0.031) is only a little higher than the observed uptake (in the range of 0.004 to 0.029), see section 3.2.1 (p9) for details.

**6.** Were clouds present during the simulation period and were they simulated? How is heterogeneous hydrolysis on cloud droplets handled?

**Response:** During the observation campaign at TMS (957 a.s.l.), there were clouds events sometimes. The $N_2O_5$ and $ClNO_2$ concentrations were below or near detection limit during such events. Therefore, we did not focus on the cloud simulation in our study. The heterogeneous processes on cloud droplets were not considered in our model.

**7.** page 7, line 3: Please add some information on the vertical model resolution. Many studies exist in the last 15 years that show pronounced gradient in N2O5 and NO3 mixing ratios, and the vertical resolution of the model is important. (e.g. Brown et al., 2007a, 2007b; Geyer and Stutz, 2004; Stutz et al., 2004, Riemer et al. 2003.)

**Response:** We used 30 model layers, and the vertical model resolution was determined using eta levels, which are shown below:

[Figure]

Figure R3. The setting of eta levels in WRF model.

The eta levels used in our study have 8 levels in the lowest 1 km or so (approximately the height of planetary boundary layer at noon), to provide more detailed information within the boundary layer.

We have added information on the vertical model resolution in the manuscript.

**8.** Table 3: Explain "Fac2"

**Response:** Fac2 is defined as the fraction of the simulations that are within a factor of two of the observations. The definition has been added to the manuscript.

**9.** Table 3 and Figure S1: It sounds like the observations of PM2.5, NO2, and O3 are available for the entire period, not only for the nights when N2O5 and ClNO2 were observed. I suggest, for figure S1, to show the entire time series, which will convey better the information if the temporal variation of the pollutant is captured. With the gaps in the time series it's hard to tell.

**Response:** The observations for the entire period at TMS along with the times series of $PM_{2.5}$, $NO_2$, and $O_3$ concentrations at environmental monitoring stations have been added to the supplement. And the temporal variations of these pollutants were not simulated as well as we had stated. We have revised the manuscript accordingly.

**10.** What is the rationale for choosing the base case for the comparison to observations in section 3.1? This seems strange to me. I would assume that the HET+Cl case is the "best effort" to capture the processes that are occurring in the real atmosphere. So, what conclusion can be

drawn from the comparison of observations to the base case? If the hydrolysis has an impact as the paper states, should we not expect a disagreement of base case and observations?

**Response:** We followed Sarwar et al. (2012) to firstly validate the performance of Base case in order to establish a reasonably good basis before we could further develop the model to include the heterogeneous chemistry of $N_2O_5$, and to evaluate the impacts of the model development.

But we agree with the reviewer that the HET+Cl case should be the 'best effort' logically, therefore, we have added model performance statistics of HET+Cl case in the manuscript. And the model performance of $O_3$ simulation in Hong Kong – Pearl River Delta region was improved in HET+Cl case, while the performance of $PM_{2.5}$ and $NO_2$ simulation did not show improvement, which could be due to many reasons, e.g. emission inventory.

**11.** page 8, line 29: calculations of averages: which hours count as "night" for the presented case?

**Response:** The time period used to calculate the nighttime average was mostly 18:00-07:00 local time, depending on the availability of the observations.

**12.** Figure 2: It would be interesting to add the "HET" case to this graph.

**Response:** The simulated $N_2O_5$ and $ClNO_2$ concentrations form 'HET' case have been added to the Figure 2 as suggested.

**13.** page 9, line 5, the statement: "the HET+Cl case captured the temporal evolution of the two compounds well". From Figure S2, I'm not sure if one can make such a statement. For some nights the peaks are roughly coinciding, for other nights not. I realize that it is very difficult to obtain good agreement with these species. There can be many reasons why there are differences between a point measurement of ClNO2 and N2O5 and a model simulation, but I'd rather suggest not making such statements in a case like this.

**Response:** We agree with the reviewer that it is a rather challenging task to well reproduce the $N_2O_5$ and $ClNO_2$ concentrations. So we have changed the statement into "the HET+Cl case generally captured the temporal variations of these two compounds."

**14.** To enhance the process-level analysis of this paper I suggest to comment on the spatial distribution of the yield $\phi$. Where in the model domain is it that ClNO2 is produced?

**Response:** The spatial plot of the simulated yield in southern China has been added to the manuscript. We have also added a comparison of the simulated yields in our study with the ones reported previously. The following sentence has been added to the manuscript.

"The simulated yield of $ClNO_2$ during night-time within PBL ranged within 0.1-0.7, which is consistent with previous observation study (0.1-0.65) (Osthoff et al., 2008) and modelling study (0-0.9) (Sarwar et al., 2012)."

**15.** The terms "under-simulated" and "over-simulated" appear frequently in the manuscript. These are not the appropriate English terms. I suggest changing this to "underpredicted" and "overpredicted".

**Response:** Corrected. Thanks for the suggestion.

**16.** page 9, line 7: the overprediction of ClNO2 can also be due to an underestimation of the sinks.

**Response:** This is a very good suggestion. We have added this possible reason. The following sentence has been added to page 9.

'Besides, the overpredicted $ClNO_2$ could also be due to the underestimation of $ClNO_2$ sink (e.g. Roberts et al, 2008).'

**17.** General comments about the figures: They are very low resolution. I suggest to submit better-quality figures for the revised version.

**Response:** Higher quality figures have been used in the revised version paper.

**18.** page 9, line 26: "within the lowest 1000 m": Does this mean that the mixing ratios were averaged over the lowest 1000 m, or is one particular layer shown in Figure 3a and c? Please clarify.

**Response:** We referred to 'the averaged mixing ratios over the lowest 8 layers (approximately 1000m)' when we used the term 'within the lowest 1000m'.

**19.** page 9, line 10: Simulated uptake coefficients higher than observed ones: From the description in section 2.2.2 it appears that organic coatings are not taken into account even though it has been shown in several studies that the presence of these can lower the uptake coefficient notably. Could the presence of organics, which is not accounted for in the simulation, explain this discrepancy and consequently also the underprediction of N2O5 and overprediction of ClNO2? Please add some discussion.

**Response:** We agree with the reviewer that the organic inhibition effect could be part of the reason that $N_2O_5$ was underpredicted and $ClNO_2$ was overpredicted.

The following sentence has been added to the manuscript "The reactive uptake coefficient could be overestimated because the parameterization used in this study (Bertram and Thornton, 2009) didn't consider the inhibition of organic coating to the uptake coefficient." in section 3.2.1 (p9).

**20.** page 10, line 5: change "suppression" to "reaction". NO3 also reacts with VOC. Does this also contribute to low NO3 concentrations near the ground?

**Response:** Corrected. The effect of VOCs to the $NO_3$ (and $N_2O_5$) concentration has been added to the manuscript.

**21.** page 10, line 9: reference to Sarwar et al (2012): Many studies have shown evidence for pronounced vertical gradients in the profiles of N2O5 before that study, see my comment 7 above.

**Response:** Several previous studies on the vertical gradients of $N_2O_5$ have been added.

**22.** page 14, line 2: "average meteorological conditions": Remind the reader what this means (average in the sense of what?)

**Response:** To demonstrate the spatial distribution of simulated $N_2O_5$ and $ClNO_2$ under general conditions or average conditions, we calculated the average concentrations of $N_2O_5$ and $ClNO_2$ in the sense of time. The word 'meteorological' in the term 'average meteorological conditions' might be a little confusing, therefore, after consideration, we deleted the word 'meteorological'.

**Reference:**

Chang, W. L., S. S. Brown, J. Stutz, A. M. Middlebrook, R. Bahreini, N. L. Wagner, W. P. Dubé, I. B. Pollack, T. B. Ryerson, and N. Riemer: Evaluating $N_2O_5$ heterogeneous hydrolysis parameterizations for CalNex 2010, J. Geophys. Res. Atmos., 121, doi:10.1002/2015JD024737, 2016.

Lowe, D., Archer-Nicholls, S., Morgan, W., Allan, J., Utembe, S., Ouyang, B., Aruffo, E., Le Breton, M., Zaveri, R.A., Di Carlo, P., Percival, C., Coe, H., Jones, R. and McFiggans, G.: WRF-Chem model predictions of the regional impacts of $N_2O_5$ heterogeneous processes on night-time chemistry over north-western Europe, Atmos. Chem. Phys., 15, 1385-1409, 2015.

Roberts, J. M., Osthoff, H. D., Brown, S. S., & Ravishankara, A. R.: N2O5 oxidizes chloride to Cl2 in acidic atmospheric aerosol, Science,321, 1059-1059, 2008,

Sarwar, G., Simon, H., Bhave, P. and Yarwood, G.: Examining the impact of heterogeneous nitryl chloride production on air quality across the United States, Atmos. Chem. Phys., 12, 6455-6473, 2012.

Wang, T., Tham, Y.J., Xue, L., Li, Q., Zha, Q., Wang, Z., Poon, S.C.N., Dubé, W.P., Blake, D.R., Louie, P.K.K., Luk, C.W.Y., Tsui, W. and Brown, S.S.: Observations of nitryl chloride and modeling its source and effect on ozone in the planetary boundary layer of southern China, J. Geophys. Res. Atmos., 121, doi: 10.1002/2015JD024556, 2016.

---

## Author Response (AR1)

**Response to Anonymous Referee #1**

We thank the referee for the comments and suggestions which help us improve the quality of the paper. Our response and the corresponding changes are listed below.

**General Comments:**

Li et al. present a set of regional photochemical modeling runs that simulate ClNO2 formation and impacts corresponding to recent field measurements in China. The field study reported the highest ever measured ambient concentration of ClNO2 indicating that this region of the world may be uniquely impacted by the chemistry associated with this compound. This study represents the first regional modeling study of ClNO2 impacts in Asia and is important for characterizing these impacts in a region with severe air pollution. The model is uniquely situated to provide a full spatial and temporal characterization of this chemistry which is not feasible with measurements alone.

One major comment is that the model performance was not great and the authors often overstate the accuracy of the performance based on the comparisons provided. Rather than gloss over the poor model performance, the authors should acknowledge this limitation and discuss how those model inaccuracies might impact their results (for instance large under-estimates of PM2.5 might lead to underestimates of the ClNO2 formation). Despite the often poor model performance, this study is valuable since it is the first application of its kind in China and provides new insights into times and locations where ClNO2 impacts are predicted to be most important. This type of characterization of spatial and temporal patterns is not possible with measurements alone.

Another major comment is that all results (figs 3-10) are given as episode averages (all hours). Since many of the pollutants modeled have distinct diurnal profiles (i.e. O3, N2O5, ClNO2) these averages are hard to interpret. For N2O5 and ClNO2, the daytime values are essentially zero so these averages include high nighttime values averaged with many zeros during daytime hours. The reader does not get a good sense of the maximum magnitude of these pollutants at night. For ozone, many areas are titrated at night so again this doesn't give any sense of how high daytime ozone values are impacted. This averaging leads the authors to make statements like "elevated levels of . . . O3 (up to 44.5 ppb)" (p 11). 44.5 ppb of ozone is generally not considered an elevated level! I suggest that the authors add some results which either show diurnal averages of changes, time series of changes, or spatial plots of max values (and maximum changes) in addition to average values. This will provide a more complete picture of the modeled impacts of this chemistry.

Overall, this analysis used the best technical information currently available to complete this modeling and I think this paper will be of interest to ACP readers. I recommend publication after the authors address general comments above and specific comments below.

**Response**: We agree with the reviewer that the model performance was not as great as we stated. We have revised the sentence describing model performance in the abstract to be 'The updated model can generally capture the temporal variation of  $N_2O_5$  and  $CINO_2$  observed at a mountaintop site in Hong Kong, but overestimates  $N_2O_5$  uptake and  $CINO_2$  production.' We also deleted the word 'satisfactorily' in section 3.1 (p8), and deleted a sentence in section 3.1 (p8) 'The capture

of the temporal variations of these pollutants at the TMS site provides a good basis for simulation of the  $N_2O_5$  and  $CINO_2$  temporal patterns (see Section 3.2.1).'

We had discussed the potential effects of the discrepancy between the simulated and observed concentrations of  $PM_{2.5}$ ,  $NO_2$  and  $O_3$  in Hong Kong – Pearl River Delta region on the  $N_2O_5$  and  $CINO_2$  simulation in section 3.2.1.

We have replaced the Figure 3 with the average  $CINO_2$  and  $N_2O_5$  concentrations during nighttime (18:00-07:00). We have added the spatial plots of average maximum values and maximum changes of NO, NO2, total nitrate and O3 in the supplement to provide extra information on the impacts of the  $N_2O_5$  and  $CINO_2$  chemistry in southern China.

**Specific comments:**

**1.** Figs 3-7, and text throughout results section: Are heights given as above ground level (agl) or above sea level (asl)? The text repeatedly says "agl" but figures indicate terrain features in black which suggests that these heights are actually "asl". Please clarify and also add text to the caption which describes the black shaded regions in the figures.

**Response**: The statement used in text, 'agl', is correct. We calculated the 'agl' based on the 'asl' and the 'terrain height' (black shaded features in cross-section figures). We have added the description of the terrain features in figures.

**2.** Page 2, Line 16: hydrolysis of NO5 is "A" major loss pathway of NOx but perhaps not "THE" major loss pathway. What about reactions of NO3 with VOCs?

**Response**: Indeed, the loss of NOx through the  $N_2O_5$  heterogeneous reaction and through the reaction between NO3 with VOCs can be both important. The word 'the' has been changed to 'a'.

**3.** Page 2, Line 26: It would be good to clarify that gamma, the reactive uptake coefficient, represents the probability that a collision between N2O5 and a particle will result in uptake and chemical reaction.

**Response**: The following explanation of the reactive uptake coefficient has been added to the manuscript 'the possibility that a colliding of  $N_2O_5$  molecule with a particle will lead to uptake and chemical reaction (Sarwar, et al. 2012).' in section 1 (p2).

**4.** Page 2, line 7 – Page 3, line 5: In the discussion of previous parameterizations for gamma, you should also mention that gamma has been measured by various field campaigns (Brown et al, 2006; Brown et al, 2009; Osthoff et al, 2008) which showed very different values for marine versus inland aerosols.

**Response**: Previous measurement studies on uptake coefficients have been added to the manuscript.

**5.** Page 3, lines 6-20: You missed several important earlier studies in your summary of modeled impacts of N2O5 and ClNO2 chemistry on air pollution concentrations/ chemistry: Dentener and Crutzen, 1993; Riemer et al., 2003; Evans and Jacob, 2005; Simon et al, 2009

**Response**: These previous studies of the impacts of N2O5 and CINO2, including Dentener and Crutzen, 1993; Riemer et al., 2003; Evans and Jacob, 2005; Simon et al, 2009, have been added to the manuscript as suggested.

6. Page 3, line 20: change "biomass burning" to "biomass burning and sea salt"

**Response: Corrected.**

**7.** Page 4, lines 18-24 and section 3.1: Comparisons with some additional measurements would be valuable if these measurements were made. For instance, if aerosol size distribution was measured that would allow for calculation of ambient surface area which could be directly compared with model results. Since surface area, not PM2.5 mass, drives this chemistry that would be a useful comparison. Also, were there any speciated PM2.5 measurements available to compare with the model (specifically aerosol nitrate and particulate chloride)? Were HCl and HNO3 measured? These could all provide better constraints and characterization of model performance if they are available. In addition, a more complete model evaluation would be useful. For instance, the authors might include time series of model performance, r2 values, maps of MB etc. There are several places in section 3.1 where the authors' characterization of the model performance is overly favorable and not supported by the figures provided. Statements that the model performed "reasonably well", "satisfactorily" etc are probably not warranted given that Figure S1 shows consistent under-predictions of ozone and PM2.5 of 20-40 ppb and 10-30 ug/m3 respectively.

**Response**: The comparison of measured and simulated surface area at TMS site has been added in section 3.1. The comparison of aerosol nitrate at TMS site has been added in section 3.1. The comparison of observed and simulated chloride had been conducted in section 3.1. No gas-phase HCl and HNO3 were measured.

Time series of the comparison of measured and simulated PM2.5, NO2, and O3 at the environmental monitoring stations and at TMS site have been added to the supplement.

We have revised the description of the model performance, see our response to the general comment.

**8.** Page 5, lines 5-15: Please specify if these are gas-phase or particle-phase chlorine emissions? Previous work on gas-phase chlorine emissions in the U.S. (Sarwar and Bhave, 2007; Chang et al, 2002) could be used as a starting point for deriving gas-phase chlorine emissions in China. Also, speciation profiles of PM2.5 emissions sources by Reff et al. (2009) could be used to derive particulate chloride emissions by applying fractional Cl contributions from all sources to the PM2.5 emissions in the current inventory. I am not suggesting that work needs to be done for this study, but the authors might discuss these past efforts as a basis for improving Chinese Cl emissions going forward.

**Response**: These are both gas and particle phase. For biomass burning, it is aerosol phase chlorine emission. For anthropogenic emission, it is gas phase chlorine (HCl) emission.

We have added a short description on the methodology that could be used to develop the chloride emission inventory in China.

**9.** Section 2.3.1: Please add information about which days were modeled. Was the modeled period Nov 15-Dec 5 to match measurements? Also, please state whether a spin-up period was included and, if so, how many days were used.

**Response**: The simulation period was November 15 to December 5, 2013, which had been stated in Section 2.3.2 (p7). The reviewer's understanding is correct. The simulation period was chosen according to the measurement period. One-day spin-up period was used.

**10.** Page 9, lines 4-5: I don't think Fig S2 supports the contention that temporal variations of N2O5 and CINO2 are "well captured". The model does predict that these pollutants build up at night and are close to 0 during daytime but other than that modeled peaks often appear at different hours and nights than observed peaks.

**Response**: We agree with the reviewer that the temporal variation of  $N_2O_5$  and  $CINO_2$  was not well captured in our simulation. The words 'well' in 'well captured' has been revised to 'generally' in section 3.2.1 (p9).

**11.** Page 9, lines 6-15: The reactive uptake coefficient could be too high in the model because the Bertram and Thornton parameterization does not account of organic inhibition of uptake that has been previously described by Riemer et al, 2009.

**Response**: We agree the reviewer's suggestion. In fact, in page 9, we had stated that in Sarwar et al. (2012), the authors attributed the parameterization (Bertram and Thornton, 2009) to be a possible reason that  $CINO_2$  was overestimated.

The following sentence has been added to the manuscript "The reactive uptake coefficient could be overestimated because the parameterization used in this study (Bertram and Thornton, 2009) does not consider the inhibition of organic coating to the uptake coefficient." in section 3.2.1 (p9).

**12.** Page 12, line 13-14: The difference between HET and HET+Cl also shows the impact of lower levels of N2O5 conversion to HNO3.

**Response**: Indeed, the difference of HET and HET+Cl, i.e. the impacts of  $CINO_2$  production, showed that less  $N_2O_5$  were transformed into  $HNO_3$  (and nitrate aerosol). And in the manuscript, we used total nitrate (HNO3+nitrate aerosol) to avoid redundant description.

**13.** Page 12, line 19: The decreases in O3 appear to only occur over a very small area, not over rural and coastal regions generally.

**Response**: We agree. The sentence has been modified accordingly.

**14.** Page 12, line 22-23: This is confusing to me. If I understand the HET run correctly, it simply set the yield value to zero for the ClNO2 pathway which should mean that more N2O5 is converted to HNO3 and less is conserved in the ClNO2 reservoir. Therefore, the HET+CL simulation should increase NOx everywhere. What would cause broad decreases in NO and NO2 across the domain with the addition of the ClNO2 formation?

**Response**: The reviewer's understanding is correct. In HET case, the ClNO2 yield was set to be zero, and all N2O5 loss was transformed into total nitrate.

The possible causes for the changes of NO and NO2 from HET to HET+Cl case are discussed below. The produced ClNO2 (1) releases NO2 and Cl radical after sunrise, and both of them increase the formation of O3. The increased O3 enhances the reaction of NO with O3, which leads to the (2) loss of NO, and the (3) formation of NO2. The increased O3 also increases the level of OH radicals, which enhances the reaction of NO2 with OH and then results in the (4) loss of NO2. The combination of the effects (1), (2), (3) and (4) mentioned above are the net effects of the ClNO2 production on NO and NO2. Our study showed that the NO was reduced across the domain. And for the NO2, the enhanced production outweighed the loss in urban areas, while in other regions, the NO2 was decreased.

We have added a short discussion of the possible causes in the manuscript.

15. Page 12, line 23: Consider rephrasing, I don't consider a decrease of 2.35 ug/m3 "slight".

**Response**: The word 'slight' has been deleted.

**16.** Page 12, line 25: What fraction of N2O5 produced ClNO2? It would be useful to report what yield values were predicted by the model for equation 5. How do these yields compare to previously reported observed yields (Osthoff et al., 2008) or modeled yields (Sarwar et al. 2012)? It might be useful to provide a map of yield values during nighttime hours.

**Response**: The spatial distribution of simulated yield during nighttime has been added to the manuscript. And the comparison of the yields with previous reported observed and simulated has been added to the manuscript. The following sentence has been added to the manuscript.

"The simulated yield of ClNO2 during night-time within PBL ranged within 0.1-0.7, which is consistent with previous observation study (0.1-0.65) (Osthoff et al., 2008) and modelling study (0-0.9) (Sarwar et al., 2012)."

**17.** Page 12, line 26-27: Simon et al. (2009) reported that half of the O3 impact from ClNO2 chemistry came from Cl activation while half came from the recycling of NO2.

Response: The conclusion from Simon et al. (2009) has been added to explain our results.

**18.** Table 3: The authors should consider including O3 performance for daytime values (8-hr daily max or 1-hr daily max) as well as all hours averages. Also what is "fac2"? This is not defined anywhere in the paper. How is it calculated?

Response: We have added 1-h daily maximum O3 as the indicator of model performance.

FAC2 is defined as the fraction of simulated results that are within a factor of two of the observations. The calculation of FAC2 is as follows.

FAC2 = (the number of simulations that are within a factor of two of the observations) / (the number of observations).

The definition of FAC2 has been added to the manuscript.

**19.** Table 4: The authors should state the time period used to calculated average simulated concentrations.

**Response**: The time period used to calculate the average simulated concentration is the entire simulation period (November 15 to December 5, 2013). It has been added to the manuscript.

**20.** Figures 4 and 5: Consider using the same scale for the horizontal and vertical plots. The concentrations don't look different enough to warrant different scales.

**Response**: The scales have been revised as suggested.

**21.** Figure 6: The choice of the log scale makes variations in the O3 concentrations harder to see. Consider using a linear scale.

**Response**: A linear scale has been applied to the O3 concentration figure.

**Response to Anonymous Referee #2**

We would like to thank the referee #2 for the comments and suggestions which help us improve the manuscript. Our response and the corresponding changes are listed below.

**General Comments:**

This paper presents WRF/Chem model simulations to assess the impacts of the heterogeneous hydrolysis of N2O5 on atmospheric chemistry for southern China, a region where high concentrations of N2O5 and ClNO2 were recently observed. A chlorine chemistry module was added to WRF/Chem to not only include HNO3 as a product of N2O5 hydrolysis, but also ClNO2, which in known to impact the oxidizing capability of the atmosphere by chlorine activation. The results show that for the chosen model domain and a simulation period during winter, N2O5 heterogeneous hydrolysis contributes significantly to the formation of particulate nitrate and ozone.

The results further point towards major model uncertainties due to chlorine emission inventories, which is consistent with previous studies. The contribution of this work consists of WRF/Chem model development and the application of the extended model to a region where, so far, not much information on the importance of N2O5 hydrolysis has been available. Obtaining good agreement between simulation and observation of N2O5 and ClNO2 is challenging, so I commend the authors for their efforts. The study fits well within the scope of ACP, and it will be of interest for the community. I recommend the paper for publication after the authors address my questions and comments below.

**Specific comments:**

1. page 2, line 25: Saer is described as aerosol surface to volume ratio. This is confusing, it should rather be the aerosol surface area density, since it refers to the aerosol surface area per volume of air.

Response: The definition of Saer has been revised to 'aerosol surface area density'.

**2.** page 3, line 22: "1-minute value", what does that exactly mean? Were you sampling every minute or averaging over many samples for 1-minute intervals?

**Response:** '1-minute value' is the average of data points collected every 6 second in a 1-minute interval. Please refer to Wang et al. (2016) for further details on the CIMS measurement.

**3.** page 6, equation 2: The factor A in this equation is a function of the surface area to volume ratio for the particles in those experiments. It would be worth checking that this is comparable to (or valid for) the study here.

**Response:** The factor 'A' in the parameterization proposed by Bertram and Thornton (2009) refers to a pre-factor which includes the ratio of the *volume to surface area* of the particles used in their experiments which is  $3.75*10^{-8}$  m, i.e. the ratio of the *surface area to volume* is  $2.67*10^{7}$  m-1.

The average simulated ratio of surface area to volume for the particles in southern China within the PBL is shown below (the interval is  $0.5*10^7 \text{ m}^{-1}$ ), from which we can see that the simulated ratio is practically within the range of 0.5 to  $2.5*10^7 \text{ m}^{-1}$  in southern China, which is very close to the value used in the parameterization.

We have added the comparison of observed and simulated ratio of the surface area to volume to the manuscript.

Figure R1. Average simulated ratio of surface area to volume (m-1) for the particles within PBL

**4.** How was the liquid water content of the aerosol determined? Are both inorganic and organic species contributing to aerosol water uptake, or is it only the inorganic species that determine the aerosol liquid water content?

**Response:** The liquid water content was predicted using the thermodynamic module, ISORROPIA. Only the inorganic species have been considered in determining the aerosol liquid water content.

**5.** Related to point 4, what is the liquid water content of the aerosols for the simulations presented here? Is the RH high enough that water uptake is predicted? For example, Lowe et al. (2015) and Chang et al. (2016) have shown that using the Bertram and Thornton parameterization can lead to problems in low RH environments — not because there is a problem with this parameterization, but rather with the way aerosol water uptake is handled in CTMs. It would be interesting to see how this study compares in this regard.

**Response:** The spatial plot of aerosol liquid water content is shown below.

Figure R2. The average aerosol liquid water content (µg m-3) in southern China during the simulation period within the PBL.

We had validated RH simulation performance, and the RH was well predicted during the simulation period (Wang et al., 2016). The mean bias between simulated and observed RH was only 3.54, and the correlation coefficient was 0.89, and the root mean square error was only 11.29.

We have noticed the conclusions of Lowe et al. (2015) and Chang et al. (2016), but it seems that our model performed relatively well to predict the uptake of  $N_2O_5$  on aerosol surface in the Hong Kong – Pearl River Delta region, given that the simulated uptake at TMS (in the range of 0.008 to 0.031) is only a little higher than the observed uptake (in the range of 0.004 to 0.029), see section 3.2.1 (p9) for details.

**6.** Were clouds present during the simulation period and were they simulated? How is heterogeneous hydrolysis on cloud droplets handled?

**Response:** During the observation campaign at TMS (957 a.s.l.), there were clouds events sometimes. The  $N_2O_5$  and ClNO2 concentrations were below or near detection limit during such events. Therefore, we did not focus on the cloud simulation in our study. The heterogeneous processes on cloud droplets were not considered in our model.

7. page 7, line 3: Please add some information on the vertical model resolution. Many studies exist in the last 15 years that show pronounced gradient in N2O5 and NO3 mixing ratios, and the vertical resolution of the model is important. (e.g. Brown et al., 2007a, 2007b; Geyer and Stutz, 2004; Stutz et al., 2004, Riemer et al. 2003.)

**Response:** We used 30 model layers, and the vertical model resolution was determined using eta levels, which are shown below:

Figure R3. The setting of eta levels in WRF model.

The eta levels used in our study have 8 levels in the lowest 1 km or so (approximately the height of planetary boundary layer at noon), to provide more detailed information within the boundary layer.

We have added information on the vertical model resolution in the manuscript.

8. Table 3: Explain "Fac2"

**Response:** Fac2 is defined as the fraction of the simulations that are within a factor of two of the observations. The definition has been added to the manuscript.

**9.** Table 3 and Figure S1: It sounds like the observations of PM2.5, NO2, and O3 are available for the entire period, not only for the nights when N2O5 and ClNO2 were observed. I suggest, for figure S1, to show the entire time series, which will convey better the information if the temporal variation of the pollutant is captured. With the gaps in the time series it's hard to tell.

**Response:** The observations for the entire period at TMS along with the times series of  $PM_{2.5}$ , NO2, and O3 concentrations at environmental monitoring stations have been added to the supplement. And the temporal variations of these pollutants were not simulated as well as we had stated. We have revised the manuscript accordingly.

**10.** What is the rationale for choosing the base case for the comparison to observations in section 3.1? This seems strange to me. I would assume that the HET+Cl case is the "best effort" to capture the processes that are occurring in the real atmosphere. So, what conclusion can be

drawn from the comparison of observations to the base case? If the hydrolysis has an impact as the paper states, should we not expect a disagreement of base case and observations?

**Response:** We followed Sarwar et al. (2012) to firstly validate the performance of Base case in order to establish a reasonably good basis before we could further develop the model to include the heterogeneous chemistry of  $N_2O_5$ , and to evaluate the impacts of the model development.

But we agree with the reviewer that the HET+Cl case should be the 'best effort' logically, therefore, we have added model performance statistics of HET+Cl case in the manuscript. And the model performance of  $O_3$  simulation in Hong Kong – Pearl River Delta region was improved in HET+Cl case, while the performance of  $PM_{2.5}$  and  $NO_2$  simulation did not show improvement, which could be due to many reasons, e.g. emission inventory.

11. page 8, line 29: calculations of averages: which hours count as "night" for the presented case?

**Response:** The time period used to calculate the nighttime average was mostly 18:00-07:00 local time, depending on the availability of the observations.

**12.** Figure 2: It would be interesting to add the "HET" case to this graph.

**Response:** The simulated  $N_2O_5$  and  $CINO_2$  concentrations form 'HET' case have been added to the Figure 2 as suggested.

**13.** page 9, line 5, the statement: "the HET+Cl case captured the temporal evolution of the two compounds well". From Figure S2, I'm not sure if one can make such a statement. For some nights the peaks are roughly coinciding, for other nights not. I realize that it is very difficult to obtain good agreement with these species. There can be many reasons why there are differences between a point measurement of ClNO2 and N2O5 and a model simulation, but I'd rather suggest not making such statements in a case like this.

**Response:** We agree with the reviewer that it is a rather challenging task to well reproduce the  $N_2O_5$  and  $CINO_2$  concentrations. So we have changed the statement into "the HET+Cl case generally captured the temporal variations of these two compounds."

14. To enhance the process-level analysis of this paper I suggest to comment on the spatial distribution of the yield  $\phi$ . Where in the model domain is it that ClNO2 is produced?

**Response:** The spatial plot of the simulated yield in southern China has been added to the manuscript. We have also added a comparison of the simulated yields in our study with the ones reported previously. The following sentence has been added to the manuscript.

"The simulated yield of ClNO2 during night-time within PBL ranged within 0.1-0.7, which is consistent with previous observation study (0.1-0.65) (Osthoff et al., 2008) and modelling study (0-0.9) (Sarwar et al., 2012)."

**15.** The terms "under-simulated" and "over-simulated" appear frequently in the manuscript. These are not the appropriate English terms. I suggest changing this to "underpredicted" and "overpredicted".

**Response:** Corrected. Thanks for the suggestion.**

16. page 9, line 7: the overprediction of CINO2 can also be due to an underestimation of the sinks.

**Response:** This is a very good suggestion. We have added this possible reason. The following sentence has been added to page 9.

'Besides, the overpredicted  $CINO_2$  could also be due to the underestimation of  $CINO_2$  sink (e.g. Roberts et al, 2008).'

**17.** General comments about the figures: They are very low resolution. I suggest to submit betterquality figures for the revised version.

**Response:** Higher quality figures have been used in the revised version paper.

**18.** page 9, line 26: "within the lowest 1000 m": Does this mean that the mixing ratios were averaged over the lowest 1000 m, or is one particular layer shown in Figure 3a and c? Please clarify.

**Response:** We referred to 'the averaged mixing ratios over the lowest 8 layers (approximately 1000m)' when we used the term 'within the lowest 1000m'.

**19.** page 9, line 10: Simulated uptake coefficients higher than observed ones: From the description in section 2.2.2 it appears that organic coatings are not taken into account even though it has been shown in several studies that the presence of these can lower the uptake coefficient notably. Could the presence of organics, which is not accounted for in the simulation, explain this discrepancy and consequently also the underprediction of N2O5 and overprediction of ClNO2? Please add some discussion.

**Response:** We agree with the reviewer that the organic inhibition effect could be part of the reason that  $N_2O_5$  was underpredicted and  $CINO_2$  was overpredicted.

The following sentence has been added to the manuscript "The reactive uptake coefficient could be overestimated because the parameterization used in this study (Bertram and Thornton, 2009) didn't consider the inhibition of organic coating to the uptake coefficient." in section 3.2.1 (p9).

**20.** page 10, line 5: change "suppression" to "reaction". NO3 also reacts with VOC. Does this also contribute to low NO3 concentrations near the ground?

**Response:** Corrected. The effect of VOCs to the  $NO_3$  (and  $N_2O_5$ ) concentration has been added to the manuscript.

**21.** page 10, line 9: reference to Sarwar et al (2012): Many studies have shown evidence for pronounced vertical gradients in the profiles of N2O5 before that study, see my comment 7 above.

**Response:** Several previous studies on the vertical gradients of N2O5 have been added.

**22.** page 14, line 2: "average meteorological conditions": Remind the reader what this means (average in the sense of what?)

**Response:** To demonstrate the spatial distribution of simulated  $N_2O_5$  and  $CINO_2$  under general conditions or average conditions, we calculated the average concentrations of  $N_2O_5$  and  $CINO_2$  in the sense of time. The word 'meteorological' in the term 'average meteorological conditions' might be a little confusing, therefore, after consideration, we deleted the word 'meteorological'.

**Reference:**

[revised manuscript text omitted]

the chemistry of  $N_2O_5/CINO_2$  in the chemical transport model, and to develop an updated chlorine emission inventory over China.

**1** Introduction**

Dinitrogen pentoxide  $(N_2O_5)$  is mostly produced by chemical reactions involving ozone  $(O_3)$  and nitrogen dioxide  $(NO_2)$ .

5

$$O_3 + NO_2 \rightarrow NO_3$$
 (1)
 $NO_3 + NO_2 \rightarrow N_2O_5$  (2)

The subsequent heterogeneous uptake of  $N_2O_5$  produces nitrate on water-containing aerosol surfaces via reaction 3 (hydrolysis) and produces both nitrate and gaseous nitryl chloride (CINO2) on chloride-containing aerosol surfaces via reaction 4 (Finlayson-Pitts et al., 1989; Osthoff et al., 2008). The net reaction of reactions 3 and 4 could be treated as reaction 5, in which the CINO2 yield, i.e. parameter  $\phi$ , represents the fraction of N2O5 that reacts via reaction 4. The produced CINO2 can be further photolysed into Cl radical and NO2 (via reaction 6).

$$\begin{split} N_2O_5(g) + H_2O(aq) &\longrightarrow 2 \text{ HNO}_3(aq) \quad (3) \\ N_2O_5(g) + \text{HCl}(aq) &\longrightarrow \text{HNO}_3(aq) + \text{ClNO}_2(g) \quad (4) \\ N_2O_5(g) + (1-\varphi)H_2O(aq) + \varphi \text{ HCl}(aq) &\longrightarrow (1-\varphi)\times 2 \text{ HNO}_3(aq) + \varphi \times (\text{HNO}_3(aq) + \text{ClNO}_2(g)) \end{split}$$

20

10

 $\operatorname{ClNO}_2(g) + hv \longrightarrow \operatorname{Cl}(g) + \operatorname{NO}_2(g)$  (6)

(5)

The above processes affect the fate and composition of the total reactive nitrogen (NOy), which is the sum of NO, NO2, HNO3 (g),  $2*N_2O_5$ , NO3, CINO2, PAN, HONO, HNO4, 
[revised manuscript text omitted]

**2.2.2 N2O5 heterogeneous uptake, CINO2 production and CI gaseous reaction**

We adopted the parameterisations of  $N_2O_5$  heterogeneous uptake and ClNO2 production suggested by Bertram and Thornton (2009). According to the parameterisations, the  $N_2O_5$  heterogeneous uptake coefficient,  $\gamma$ , can be calculated with the following equation:

$$\gamma = Ak(1 - \frac{1}{\frac{(0.06[H_2O(1)])}{[NO_3]} + 1 + (\frac{29[C1^-]}{[NO_3]})})$$
(Equation 2)

where  $A=3.2\times10^{-8}$ ,  $k=1.15\times10^{6}\times(1-e^{(-0.13[H_2O(1)])})$ , and  $[H_2O(1)]$ ,  $[NO_3^-]$  and  $[Cl^-]$  are the molarities of liquid water, nitrate, and chloride in aerosol volume. The yield of ClNO2,  $\phi$ , can be calculated with the following equation:

$$\phi = (1 + \frac{[H_2O(1)]}{483[C1^-]})^{-1}$$
 (Equation 3).

The loss of N2O5 and the production of nitrate and ClNO2 can be predicted with Eq. (1-3). The produced ClNO2 
[revised manuscript text omitted]

**3.2 Simulation of N2O5 and CINO2 with N2O5 uptake and Cl activation**

**3.2.1 Comparison of simulated N2O5 and ClNO2 with observation**

- The average observed and simulated (HET+Cl case) concentrations of  $N_2O_5$  and ClNO2 were calculated for each night, as 25 shown in Fig. 2. The mean observed  $N_2O_5$  concentrations for each night varied from 0.02 to 0.74 ppb during the study period, while the average simulated  $N_2O_5$  values from the HET+Cl case were between 0.02 and 0.35 ppb. The HET+Cl case reproduced the order of  $N_2O_5$  concentrations but underestimated them within a factor of three. Differences between the HET case and HET+Cl case in the simulated N2O5 were unnoticeable. For ClNO2, the average observed concentrations varied from 0.01 to 0.39 ppb, whilst the mean simulated values for each night varied between 0.05 and 0.42 ppb. The HET+Cl case reproduced the order of CINO2 concentrations with an overestimate mostly within a factor of four. The simulated and observed
- 30

hourly concentrations of  $N_2O_5$  and ClNO2 are shown in Fig. S2S3, indicating that the HET and HET+Cl case well-captured the temporal variations of these two compounds in general.

The under simulated  $N_2O_5$  and over simulated  $O_2$  values in the HET+Cl case point to the underestimation of the sources and/or the overestimation of the sink of  $N_2O_5$  and the overestimation of the production of ClNO2.

- As shown in section 3.1, the simulated NO2 and O3 levels in the HK-PRD region are lower than the observations, which results in lower production of N2O5; the simulated PM2.5 concentrations are higher than the observed values which would lead to an overestimate of N2O5 heterogeneous loss. The observation-derived N2O5 uptake coefficients at the TMS site (Brown et al., 2016) varied from 0.004 to 0.029 with an average value of 0.014, whilst the simulated uptake coefficients ranged from 0.008 to 0.031 with an average of 0.019, which suggests that the HET+Cl simulation generally overestimates N2O5 uptake coefficients, which causes further overestimation of the loss of N2O5. The reactive uptake coefficient could be overestimated because the parameterization used in this study does not consider the inhibition of organic coating to the uptake coefficient (Bertram and Thornton, 2009). The overestimated loss of N2O5 on aerosol inherently overestimated the production of ClNO2. The parameterisations used in this study are likely to overestimate the ClNO2 yield (Kim et al., 2014; Ryder et al., 2015), which would further overestimate the production of ClNO2. In addition, the overpredicted ClNO2 could also be due to the
- 15 ignorance of possible CINO2 sinks (e.g. Roberts et al, 2008).

Discrepancies between the measured and simulated  $N_2O_5$  and CINO2 levels have also been reported in previous model studies. Lowe et al. (2015) used the same parameterisations for  $N_2O_5$  uptake that we applied in our study and showed slightly higher average simulated  $N_2O_5$  values along two flight tracks but a factor of 1-2 lower simulated  $N_2O_5$  in another flight. They noted that the underestimated  $N_2O_5$  could be attributed to inaccuracies in the meteorological simulation. Sar